# Farzi Data: Autoregressive Data Distillation

## Abstract

We study data distillation for auto-regressive machine learning tasks, where the input and output have a strict left-to-right causal structure. More specifically, we propose Farzi, which summarizes an event sequence dataset into a small number of *synthetic* sequences — Farzi Data — which are optimized to maintain (if not improve) model performance compared to training on the full dataset. Under the hood, Farzi conducts memory-efficient data distillation by (i) deriving efficient reverse-mode differentiation of the Adam optimizer by leveraging Hessian-Vector Products; and (ii) factorizing the high-dimensional discrete event-space into a latent-space which provably promotes implicit regularization. Empirically, for sequential recommendation and language modeling tasks, we are able to achieve $98 - 120\%$ of downstream full-data performance when training state-of-the-art models on Farzi Data of size as little as $0.1\%$ of the original dataset. Notably, being able to train *better models with significantly less data* sheds light on the design of future large auto-regressive models, and opens up new opportunities to further scale up model and data sizes.

## 1 Introduction

The effectiveness of machine learning models relies heavily on the *quantity* and *quality* of training data. While the quantity of training data is always well-regarded in the scaling-laws of training highly-parameterized neural networks (Hoffmann et al., 2022; Kaplan et al., 2020; Borgeaud et al., 2022; Zhai et al., 2022; Du et al., 2022), the quality of underlying data is often overlooked. Despite being an intuitive covariate in downstream model performance, there does not exist an efficient out-of-the-box solution for measuring the quality of a data point. Some popular heuristics (*e.g.*, data valuation (Ghorbani & Zou, 2019), coresets (Borsos et al., 2020a)) fall short from a variety of angles (Basu et al., 2021; Kumar et al., 2020; Toneva et al., 2019; Sener & Savarese, 2018).

Data distillation (DD) (see Sachdeva & McAuley (2023) for a comprehensive survey) offers a promising alternative to explicitly tagging the quality of each datapoint. Loosely, DD approaches aim to *synthesize* a terse data summary solely intended to train models to the same (if not better) quality as training them on the original dataset. In this paper, we propose Farzi, a DD approach designed specifically for synthesizing high-fidelity auto-regressive data summaries. We call the data synthesized by Farzi as Farzi Data.

Farzi Data takes a step towards addressing the massive costs (*e.g.*, financial, environmental, *etc.*) associated with training large auto-regressive models (OpenAI, 2023; Anil et al., 2023; Radford et al., 2022) on massive amounts of pretraining data by (i) implicitly *filtering* out low-quality sources of information resulting in a terse data summary, and (ii) *re-organizing* the data in a format that is most pertinent for model training. Intuitively, a vast majority of underlying *information* in such auto-regressive datasets is redundant from the downstream task's perspective. For example, looking at recommender systems, a predictive model wouldn't necessarily need trillions of event-level data from billions of users to accurately model user-behaviour patterns.

Typical DD techniques (Zhao et al., 2021; Zhao & Bilen, 2023; 2021; Nguyen et al., 2021; Cazenavette et al., 2022; Zhou et al., 2022b; Deng & Russakovsky, 2022) are geared toward low-resolution image datasets due to (i) computationally expensive data optimization, and (ii) generation-friendly continuous domain of images (pixels). On the other hand, auto-regressive data generally consists of sequences of discrete *tokens* (*e.g.*, sub-words, item-IDs, *etc.*) with a potentially large vocabulary. Further, in many applications, each sequence can entail a long list of such tokens. Farzi addresses

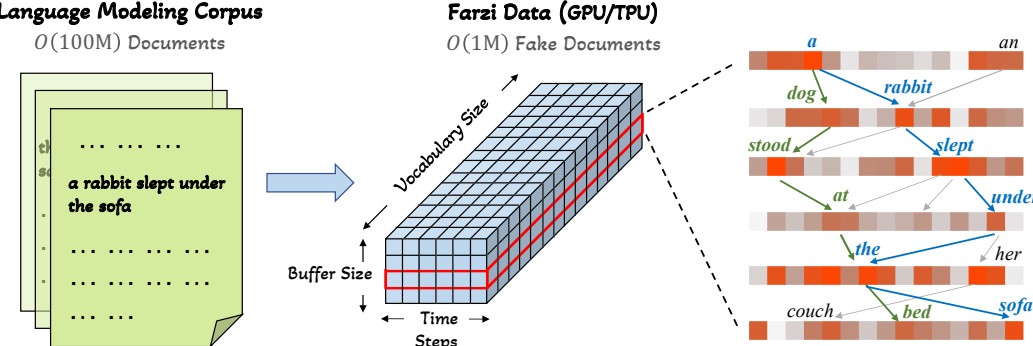

Figure 1: Visualization of FARZI DATA in the context of language modeling. FARZI DATA can be seen as a 3-D tensor comprising of *sequences of distributions over tokens*, where a single *distribution sequence* fuses the information content of multiple discrete sequences. E.g, a single distribution sentence can unfold into an entire tree of similar sentences like "the rabbit slept under the sofa" or "a dog stood at the bed" as depicted in the figure. Such a parameterization: (i) makes the dataset much more GPU/TPU friendly; (ii) reduces the cardinality of the dataset leading to efficient training; and (iii) enables models to be trained on *fuzzy sequences*, hopefully leading to robust learning.

the aforementioned characteristics of auto-regressive data by performing *data distillation in a latent space* by organizing FARZI DATA into (i) a *latent* data summary that captures the downstream task patterns, and (ii) a decoder (*e.g.*, token-embeddings) that maps the latent-space back to the token-space. In addition to making FARZI optimization-friendly (both of the aforementioned data components are non-discrete/continuous), we demonstrate that such latent-parameterization provably promotes implicit regularization when training models on FARZI DATA (Theorem 3.1).

We highlight four main contributions of this paper:

- We develop FARZI, a scalable `DD` technique for summarizing massive auto-regressive datasets, and demonstrate FARZI DATA's sample efficiency over 5 datasets spanning sequential recommendation and language modeling tasks. Training on FARZI DATA, we are able to achieve up to $98 - 120\%$ of full-data performance for state-of-the-art models using as little as $0.1\%$ of the original dataset size, as well as noting a strong cross-architecture generalization, *i.e.*, being able to train various (student) models on FARZI DATA synthesized using a given (teacher) model.

- Building atop the meta-matching framework of `DD` (see Sachdeva & McAuley (2023) for a taxonomy), we propose two crucial modifications for largely improved sample efficiency. First, conducting an investigative study on the role of inner-loop optimizer in `DD`, we conclude Adam (Kingma & Ba, 2015) to be much more adept than SGD (with or without momentum) for `DD`. This is in stark contrast with existing `DD` and meta-learning studies where SGD is the de-facto optimizer of choice. We further improve FARZI's sample quality by leveraging pretrained training trajectories for initialization in the meta-matching optimization.

- In addition to generating high-fidelity data, FARZI is computationally highly scalable. Firstly, parameterizing FARZI DATA into a latent data summary and a token decoder saves large amount of time and memory during optimization, thereby making FARZI (roughly) independent of the vocabulary size. Further, we derive an efficient reverse-mode differentiation of Adam which has a memory complexity independent of the number of inner-loop steps, unlike autograd systems which store all intermediate variables, therefore leading to $\mathcal{O}(100)\times$ memory footprint reduction.

- We provide a formal analysis of FARZI from various standpoints. We firstly show that FARZI DATA's latent parameterization implicitly promotes regularization and provably improves generalization. Previous studies have observed such *data overfitting* effects in `DD` empirically (Zhou et al., 2022b), but we are the first to study its theoretical underpinnings. We further demonstrate the correctness of our proposed reverse-mode differentiation of Adam.

## 2 RELATED WORK

**Data downsampling.** The complexity and training time for state-of-the-art models from different domains has grown exponentially in the recent years (OpenAI, 2023; Sun et al., 2019; Mittal et al., 2021; Rombach et al., 2022). Sampling has been the classic approach to summarize large datasets, approaches for which can be grouped into the following categories: **(i) Coreset construction** techniques which sample a weighted subset of the given dataset to accelerate model training (Kaushal et al., 2019; Borsos et al., 2020b; Krause et al., 2021; Kazemi et al., 2021). Being a combinatorial optimization, coreset construction techniques typically leverage submodularity assumptions (Bilmes, 2022) to optimize the coreset in a tractable manner. **(ii) Data valuation** approaches which typically leverage shapley values (Shapley, 1953) to tag the *value* of each data point for model training (Wang & Jia, 2023; Ghorbani & Zou, 2019; Kwon & Zou, 2023; Kwon et al., 2021). Notably, such data valuation methods turn out to be computationally intractable even for moderate sized datasets. **(iii) Heuristic samplers** that build upon designing ad-hoc notions of data quality. Two prominent schools-of-thought in designing such heuristics has been to either preserve notions like diversity (Coleman et al., 2022; Abbas et al., 2023; Sorscher et al., 2022), discrepancy (Karnin & Liberty, 2019), *etc.* in some metric-space of the inputs, or use the loss-values from some proxy model to tag the difficulty (and thereby, quality) for each datapoint (Paul et al., 2021; Coleman et al., 2020; Sachdeva et al., 2021; Jiang et al., 2019).

**Data distillation.** Contrary to sampling datapoints from a given dataset, data distillation approaches aim to *synthesize* high-quality data summaries for sample-efficient model training through bilevel optimization (see Sachdeva & McAuley (2023) for a comprehensive survey). Prominent existing approaches are designed for summarizing images (Wang et al., 2018; Zhao et al., 2021; Zhao & Bilen, 2023; 2021; Cazenavette et al., 2022; Zhou et al., 2022b; Deng & Russakovsky, 2022; Nguyen et al., 2021), graphs (Jin et al., 2022a;b), and recommender systems (Sachdeva et al., 2022a). Such approaches can essentially be viewed as meta-learning approaches (see Hospedales et al. (2021) for a comprehensive survey) with the meta-optimization happening over the data summary instead of common applications like model initialization (Finn et al., 2017) or task hyper-parameters (Maclaurin et al., 2015; Lorraine et al., 2020).

**Autoregressive tasks.** A variety of machine learning tasks are auto-regressive, *e.g.*, language modeling (OpenAI, 2023; Gokaslan et al., 2019; Raffel et al., 2019), sequential recommendation (Sachdeva et al., 2019; Kang & McAuley, 2018; Bennett et al., 2007), self-driving (Sachdeva et al., 2022b; Sun et al., 2020), *etc.* Such tasks have a clear left-to-right causal structure with one event preceding the other, typically in time. Further, since a majority of such tasks are semi-supervised and are associated with large-amounts of naturally occurring data; training large foundation models (Bommasani et al., 2021) for such data can become daunting despite its practicality, thereby limiting overall research progress. Concerningly, to the best of our knowledge, only simple *data sampling heuristics* scale to such large auto-regressive datasets (Toneva et al., 2019; Sener & Savarese, 2018).

## 3 FARZI: SYNTHESIZING HIGH-FIDELITY AUTOREGRESSIVE DATA SUMMARIES

**Task & Notation.** Given an autoregressive dataset $\mathcal{D} \triangleq \{\mathbf{x}_i\}_{i=1}^{|\mathcal{D}|}$ where $\mathbf{x}_i \triangleq [x_{ij} \in \mathcal{V}]_{j=1}^{|\mathbf{x}_i|}$ is an ordered sequence of tokens, each belonging to the vocabulary of all possible tokens $\mathcal{V}$. We aim to synthesize a data summary $\mathcal{D}_{\mathsf{syn}} \in \mathbb{R}^{\mu \times \xi \times \dim(\mathcal{V})}$ consisting of $\mu$ fake sequences of maximum length $\xi$, *s.t.*, $\mu \ll |\mathcal{D}|$. More specifically, we seek to construct $\mathcal{D}_{\mathsf{syn}}$ in such a way that a representative learning algorithm $\Phi_\theta : \mathcal{V}^n \mapsto \mathcal{V}$ trained on $\mathcal{D}_{\mathsf{syn}}$ using an autoregressive task (*e.g.*, next-token-prediction (Radford et al., 2018), cloze (Taylor, 1953), *etc.*) specified by a cost function $l : \mathcal{V} \times \mathcal{V} \mapsto \mathbb{R}$ can achieve performance equivalent to that of training $\Phi_\theta$ on the original dataset $\mathcal{D}$. Taking next-token-prediction (Radford et al., 2018) as a representative predictive task, we denote the empirical risk as $\mathcal{L}_\mathcal{D}(\theta) \triangleq \mathbb{E}_{\mathbf{x} \sim \mathcal{D}, \, x_i \sim \mathbf{x}}[l(\Phi_\theta(\mathbf{x}_{1:i}), x_{i+1})]$ for notational convenience, where $\mathbf{x}_{1:i}$ represents the sequence of first $i$ tokens in $\mathbf{x}$.

**Methodology.** We cast the problem of autoregressive DD as a meta-learning problem, wherein the *inner-loop* trains a learning algorithm on the data summary, and the *outer-loop* evaluates its *quality*

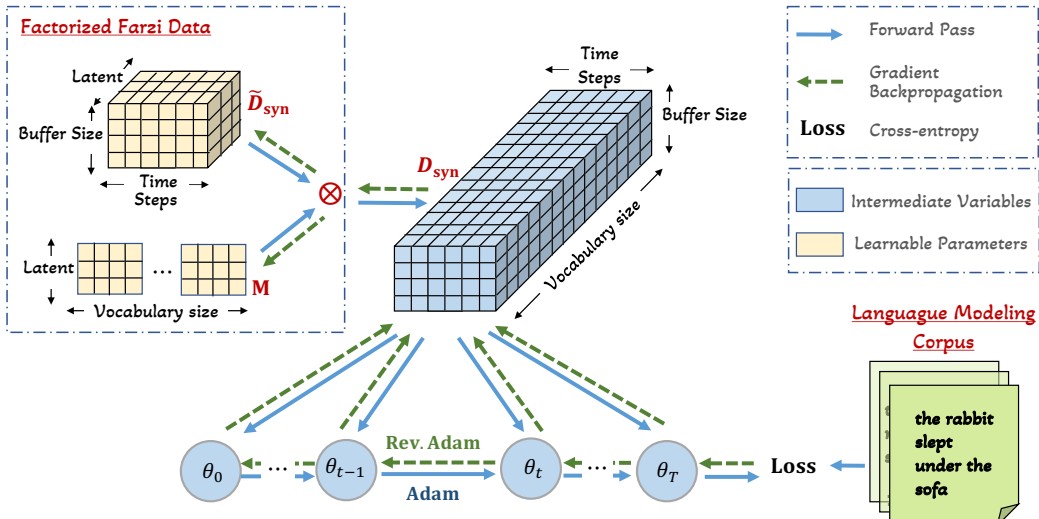

Figure 2: Visualization of a single outer-loop step in FARZI demonstrated using the language modeling predictive task. In this framework, each outer-loop step first materializes FARZI DATA using (a batch of) its respective low-rank counterparts, followed by training a learning algorithm on FARZI DATA for $T$-steps using Adam. The meta-gradient to update the factorized FARZI DATA is obtained using efficient reverse-mode Adam outlined in Algorithm 1. This process (outer-loop step) is repeated till convergence, or for a fixed number of iterations.

via $l(\cdot, \cdot)$ on the original dataset to directly update the data summary via gradient descent. More formally, a naïve bilevel optimization problem can be framed as follows:

$$\arg\min_{\mathcal{D}_{\text{syn}}} \; \mathbb{E}_{\theta_0 \sim \Theta} \left[ \mathcal{L}_{\mathcal{D}}(\theta^*) \right] \quad \text{s.t.} \quad \theta^* \triangleq \arg\min_{\theta} \; \mathcal{L}_{\mathcal{D}_{\text{syn}}}(\theta \mid \theta_0) \;, \tag{1}$$

where $\Theta$ is a distribution to initialize model parameters (*e.g.*, uniform, Kaiming (He et al., 2015), *etc.*). Such a formulation is commonly termed as *meta-model matching based DD* (see Sachdeva & McAuley (2023) for a taxonomy of existing approaches), and is associated with significant computational complexity in terms of both time and memory. Typical approaches resort to local optimization (*e.g.*, SGD) in the inner-loop, and Truncated Backpropagation Through Time (T-BPTT) by unrolling a finite number of inner optimization steps to obtain the meta-gradient. Notably, DD becomes infeasible — even after making such assumptions — when the data is autoregressive as each data-point is associated with (i) a large discrete token vocabulary, *i.e.*, $\dim(\mathcal{V})$; and (ii) a third sequential dimension, *i.e.*, $\xi$. Hence, the computational complexities of existing DD techniques grows by a factor of $\approx \xi \cdot \dim(\mathcal{V})$.

To alleviate the computational challenges, FARZI performs *data distillation in a latent space*. More specifically, FARZI factorizes $\mathcal{D}_{\text{syn}}$ into: (i) a latent data summary $\tilde{\mathcal{D}}_{\text{syn}} \in \mathbb{R}^{\mu \times \xi \times d}$ where $d \ll \dim(\mathcal{V})$; and (ii) a token-decoder matrix $\mathbf{M} \in \mathbb{R}^{d \times \dim(\mathcal{V})}$. Finally, we can compose the latent data summary and the token-decoder to obtain the final data summary: $\mathcal{D}_{\text{syn}} \equiv \text{softmax}(\tilde{\mathcal{D}}_{\text{syn}} \cdot \mathbf{M} / \tau)$, where $\tau \in \mathbb{R}^+$ represents the temperature in $\text{softmax}(\cdot)$ and controls the entropy in $\mathcal{D}_{\text{syn}}$. Such a factorization makes FARZI scalable to both extremely large datasets, *i.e.*, large $|\mathcal{D}|$ as well as datasets with large token vocabularies, *i.e.*, large $\dim(\mathcal{V})$.

In addition to promoting scalability, we prove that FARZI DATA's latent parameterization implicitly promotes regularization while training downstream models (Theorem 3.1). More specifically, we leverage the concepts of *data representativeness* and *Rademacher complexities* (Shalev-Shwartz & Ben-David, 2014, Chapter 26) to show that explicit rank regularization while synthesizing data summaries (*e.g.*, latent factorization) strictly promotes generalization. Notably such *data overfitting* has been previously (empirically) noted to notoriously affect DD (Zhou et al., 2022b), but we are the first to explore the theoretical underpinnings.

---

**Algorithm 1** Reverse-mode differentiation of Adam. See Appendix A for the Adam algorithm.

1: **Input:** $\mathbf{w}_T, \mathbf{m}_T, \mathbf{v}_T, \gamma, \alpha, \epsilon, L(w, x)$, meta-objective $f(w)$
2: **Initialize:** $d\mathbf{m} \leftarrow 0, d\mathbf{x} \leftarrow 0, d\mathbf{w} \leftarrow \nabla_\mathbf{w} f(\mathbf{w}_T)$
3: **for** $t = T$ **to** $1$ **do**
4:    $\hat{\mathbf{m}}_t \triangleq \mathbf{m}_t / (1 - \beta_1^t)$                          ▷ exactly reverse Adam
5:    $\hat{\mathbf{v}}_t \triangleq \mathbf{v}_t / (1 - \beta_2^t)$                          ▷ exactly reverse Adam
6:    $\mathbf{w}_{t-1} = \mathbf{w}_t + \alpha \cdot \hat{\mathbf{m}}_t / (\hat{\mathbf{v}}_t + \epsilon)$                          ▷ exactly reverse Adam
7:    $\mathbf{g}_t \triangleq \nabla_\mathbf{w} L(\mathbf{w}_{t-1}, \mathbf{x})$                          ▷ exactly reverse Adam
8:    $\mathbf{m}_{t-1} = [\mathbf{m}_t - (1 - \beta_1) \cdot \mathbf{g}_t] / \beta_1$                          ▷ exactly reverse Adam
9:    $\mathbf{v}_{t-1} = [\mathbf{v}_t - (1 - \beta_2) \cdot \mathbf{g}_t^2] / \beta_2$                          ▷ exactly reverse Adam
10:    $\epsilon' \triangleq \epsilon \cdot \sqrt{1 - \beta_2^t}$
11:    $\alpha' \triangleq \alpha \cdot \sqrt{1 - \beta_2^t} / (1 - \beta_1^t)$
12:    $\beta' \triangleq (1 - \beta_2) / (1 - \beta_1)$
13:    $d\mathbf{m} = d\mathbf{m} + \alpha' \cdot \left( \frac{\beta' \cdot \mathbf{m}_t \cdot \mathbf{g}_t}{\sqrt{\mathbf{v}_t} \cdot (\sqrt{\mathbf{v}_t} + \epsilon')^2} - \frac{1}{\sqrt{\mathbf{v}_t} + \epsilon'} \right) \cdot d\mathbf{w}$                          ▷ Proposition 3.2
14:    $d\mathbf{w} = d\mathbf{w} - (1 - \beta_1) \cdot d\mathbf{m} \cdot \nabla_\mathbf{w} \nabla_\mathbf{w} L(\mathbf{w}_{t-1}, \mathbf{x})$                          ▷ Hessian-vector product
15:    $d\mathbf{x} = d\mathbf{x} - (1 - \beta_1) \cdot d\mathbf{m} \cdot \nabla_\mathbf{x} \nabla_\mathbf{w} L(\mathbf{w}_{t-1}, \mathbf{x})$                          ▷ Hessian-vector product
16:    $d\mathbf{m} = \beta_1 \cdot d\mathbf{m}$
17: **Output:** gradient of $f(\mathbf{w}_T)$ *w.r.t.* $\mathbf{w}_0, \mathbf{m}_0$, and $\mathbf{x}$

---

**Theorem 3.1.** *Let $\mathcal{D}_{\mathsf{syn}} \in \mathbb{R}^{\mu \times \xi \times \dim(\mathcal{V})}$ be parameterized using $\tilde{\mathcal{D}}_{\mathsf{syn}} \in \mathbb{R}^{\mu \times \xi \times d}$ and $\mathbf{M} \in \mathbb{R}^{d \times \dim(\mathcal{V})}$, and $\mathcal{D}_{\mathsf{naive}} \in \mathbb{R}^{\mu \times \xi \times \dim(\mathcal{V})}$ denote the non-parameterized data. Let $\mathcal{F}$ be the function-class of quadratic classifiers, and $\mathrm{Rep}(\mathcal{F}, \mathcal{D})$ denote the representativeness of a training set $\mathcal{D}$ (lower is better); then if $d < \min(\mu, \xi \cdot \dim(\mathcal{V}))$:*

$$\mathbb{E}_{\tilde{\mathcal{D}}_{\mathsf{syn}}, \mathbf{M}}[\mathrm{Rep}(\mathcal{F}, \tilde{\mathcal{D}}_{\mathsf{syn}} \cdot \mathbf{M})] < \mathbb{E}_{\mathcal{D}_{\mathsf{naive}}}[\mathrm{Rep}(\mathcal{F}, \mathcal{D}_{\mathsf{naive}})] \ .$$

*Proof.* See Appendix B.1 for the relevant preliminaries and proof. □

While typical bilevel optimization approaches use SGD in the inner loop (Deng & Russakovsky, 2022) due to efficient reversible dynamics of SGD (see Maclaurin et al. (2015) for efficient reverse-mode SGD), we empirically observe that in our setting of autoregressive DD, Adam optimization (Kingma & Ba, 2015) in the inner-loop is crucial for downstream DD performance (see Figure 5). Further, we also note that a significant number of inner-loop optimization steps — in the order of 100s — are needed for good generalization for both Adam and SGD based DD, as is concurrently reported by other work (Deng & Russakovsky, 2022). To this end, we derive an efficient approximation of reverse-mode differentiation of the Adam optimization in Algorithm 1.

**Proposition 3.2.** *Correctness of Algorithm 1, Line 13 : see Appendix B.2 for the proof.*

Algorithm 1 allows the memory footprint of the meta-gradient computation to be constant *w.r.t.* the number of inner-loop steps. Notably, meta-gradient computation is the biggest contributor in a meta-learning algorithm's overall scalability. This is in stark contrast with typical autograd libraries like PyTorch (Paszke et al., 2019), JAX (Bradbury et al., 2018), *etc.* which require storing all intermediate variables across the inner-optimization to compute the meta-gradient, resulting in a linearly growing memory footprint *w.r.t.* the number of inner-loop steps.

FARZI also improves the sample-efficiency of the underlying meta-matching framework (Equation (1)) by leveraging access to a limited number of training trajectories on the target dataset. Formally, let $\Omega \triangleq \{[\theta_i]_{i=1}^T \mid \theta_0 \sim \Theta\}$ be the set of episodic checkpoints of training $\Phi_\theta$ on $\mathcal{D}$ for a limited number of random initializations. FARZI leverages $\Omega$ in its final optimization as follows:

$$\underset{\mathbf{M}, \tilde{\mathcal{D}}_{\mathsf{syn}}}{\arg\min} \ \underset{\theta_0 \sim \Omega}{\mathbb{E}} [\mathcal{L}_\mathcal{D}(\theta_T)] \quad \text{s.t.} \quad \theta_{t+1} \leftarrow \mathrm{Adam}\left(\theta_t, \nabla_\theta \mathcal{L}_{\mathcal{D}_{\mathsf{syn}}}(\theta_t)\right)$$

$$\mathcal{D}_{\mathsf{syn}} \leftarrow \mathrm{softmax}\left(\tilde{\mathcal{D}}_{\mathsf{syn}} \cdot \mathbf{M} / \tau\right) , \tag{2}$$

where $\mathrm{Adam}(\cdot, \cdot)$ represents the set of Adam update equations listed in Appendix A, and $T$ represents the number of inner-loop optimization steps for each outer-loop step. Notably, curating $\Omega$ is independent of the DD procedure and can be precomputed and logged beforehand, contributing nothing to the computational complexity of FARZI.

**Computational complexity.** We elucidate FARZI's computational footprint of optimizing Equation (2) in terms of a single outer-loop step's runtime and memory usage:

$$
\text{Memory Complexity:} \quad \mathcal{O}\Big( \underbrace{|\Phi|}_{\text{Model optimization}} + \underbrace{b \cdot \dim(\mathcal{V})}_{\text{Storing } \hat{\mathcal{D}}} + \underbrace{b_{\mathsf{syn}} \cdot \xi \cdot \dim(\mathcal{V})}_{\text{Storing } \hat{\mathcal{D}}_{\mathsf{syn}}} + \underbrace{\mu \cdot \xi \cdot d}_{\text{Storing \& Updating } \tilde{\mathcal{D}}_{\mathsf{syn}}} + \underbrace{d \cdot \dim(\mathcal{V})}_{\text{Storing \& Updating } \mathbf{M}} \Big)
$$

$$
\text{Time Complexity:} \quad \mathcal{O}\Big( \underbrace{b_{\mathsf{syn}} \cdot \xi \cdot d \cdot \dim(\mathcal{V})}_{\text{Computing } \hat{\mathcal{D}}_{\mathsf{syn}} \text{ from } \mathcal{D}_{\mathsf{syn}} \, \& \, \mathbf{M}} + \underbrace{T \cdot b_{\mathsf{syn}} \cdot |\Phi|}_{\text{Inner-loop optimization}} + \underbrace{b \cdot |\Phi|}_{\text{Computing } \nabla_\theta \mathcal{L}_{\hat{\mathcal{D}}}(\theta_T)} +
$$
$$
\underbrace{T}_{\text{Inner-loop reversal (Algorithm 1)}} \cdot ( \underbrace{b_{\mathsf{syn}} \cdot |\Phi|}_{\text{Computing } \nabla_\theta \mathcal{L}_{\hat{\mathcal{D}}_{\mathsf{syn}}}(\theta_t)} + \underbrace{b_{\mathsf{syn}} \cdot \xi \cdot d}_{\text{Updating meta-gradient for } \tilde{\mathcal{D}}_{\mathsf{syn}}} + \underbrace{d \cdot \dim(\mathcal{V})}_{\text{Updating meta-gradient for } \mathbf{M}} ))\Big)
$$

where, $\hat{\mathcal{D}} \sim \mathcal{D}$ and $\hat{\mathcal{D}}_{\mathsf{syn}} \sim \mathcal{D}_{\mathsf{syn}}$ are randomly sampled batches of real data and FARZI DATA such that $b \triangleq |\hat{\mathcal{D}}|$ and $b_{\mathsf{syn}} \triangleq |\hat{\mathcal{D}}_{\mathsf{syn}}|$; and $|\Phi|$ represents the total number of parameters in $\Phi$.

## 4 EMPIRICAL EVALUATION

### 4.1 SETUP

We empirically evaluate FARZI's practicality over two well-studied autoregressive predictive tasks:

- *Sequential Recommendation:* Predict the *item* that a given user is most likely to consume next, given their historic item consumption history. We use four benchmark datasets, namely Movielens-100k, Movielens-1M (Harper & Konstan, 2015), Amazon Magazine (Ni et al., 2019a), and Netflix (Bennett et al., 2007); from different recommendation domains and with varying data characteristics. To evaluate model quality we use popular ranking metrics: AUC, HitRate, and nDCG. A detailed description of all datasets and metrics can be found in Appendices C.1 and C.2.

- *Language Modeling (LM):* Predict the most probable following word given a sequence of words. We conduct our experiments on the official-released train/validation/test split of the English Penn Treebank (PTB) corpus (Marcus et al., 1993): an open-sourced benchmark widely used for LM. We evaluate our models using word-level perplexity, as well as the token prediction accuracy after greedy decoding on the test set. Further details about the dataset and metrics are described in Appendices C.1 and C.2.

We use SASRec (Kang & McAuley, 2018) and a small Transformer model (Vaswani et al., 2017) as the representative learning algorithms ($\Phi$) in FARZI's inner-loop for sequential recommendation and language modeling tasks respectively, and use cross-entropy as the underlying objective function for both. We implement FARZI using PyTorch (Paszke et al., 2019) and we will publicly release the code and optimized FARZI DATA for all datasets used in this paper upon acceptance. We conduct all our experiments on a single RTX 2080-Ti GPU (11 GB), and list all relevant hyper-parameters and further experimental details in Appendices C.3 and C.4.

### 4.2 EXPERIMENTS

**How sample efficient is FARZI DATA?** We evaluate the fidelity of FARZI DATA by first optimizing for $\mathcal{D}_{\mathsf{syn}}$ using Equation (2), followed by training $\Phi$ (from scratch) on $\mathcal{D}_{\mathsf{syn}}$. We plot the performance of the trained $\Phi$ on the test-set for various amounts of data budgets ($\mu$) in Figures 3 and 4 for sequential recommendation and language modeling tasks respectively. A tabular version of the same results can be found in Appendix D, Table 6. We also plot semantically equivalent results for other

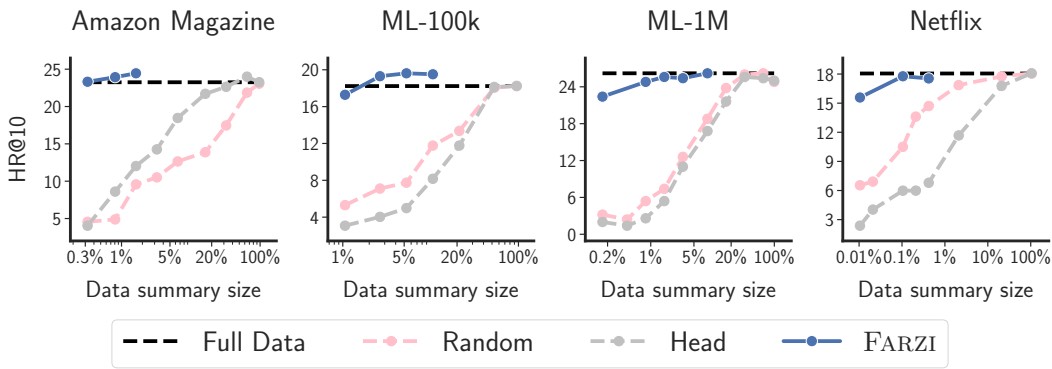

Figure 3: Performance change of SASRec with increasing data (log-scale) for recommendation. For a tabular version of these results see Appendix D, Table 6; and for results on more metrics see Appendix D, Figure 9.

commonly used data sampling heuristics, namely (i) random sampling: sample sequences uniformly at random, and (ii) head sampling: retain the sequences with the largest length. We first note that FARZI DATA is much more sample-efficient than other data sampling techniques, being able to achieve up to $1000\times$ data compression with no loss in performance. Further, in Figure 3, we notice that on two out of the four recommendation datasets, FARZI's orders of magnitude smaller data is able to train models of higher quality than the original dataset itself. This observation acts as further evidence for the intuitive yet under-explored idea that less but high-quality data can be more important for model training than a very large but noisy dataset (Sachdeva et al., 2022a; Zhou et al., 2023).

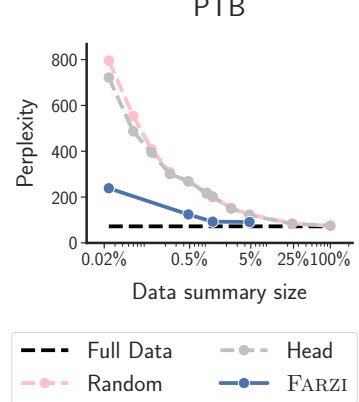

Figure 4: Performance change of Transformer with increasing data for LM.

**How versatile is FARZI DATA?** Since FARZI DATA is inherently optimized for a specific learning algorithm, we ascertain its universality by training different kinds of `student networks` over data synthesized using a given `teacher network` in FARZI's inner-loop for the sequential recommendation task. Note that the `student network` is completely unrelated to the data synthesis procedure and underlying FARZI optimization. From the results in Table 1, we observe that irrespective of the `teacher network`, FARZI DATA is able to train varied `student network` architectures (*e.g.*, Transformers, RNNs, MLPs) better than training on the full dataset. On the other hand, however, the best performance for any given `student network` is obtained when the same network is used during FARZI optimization.

**How important is the inner-loop optimizer in FARZI?** We compare SGD (with or without momentum) and Adam (Kingma & Ba, 2015) optimizers as different optimization routines in FARZI's inner-loop (Equation (2)). Notably, we implement differentiable Adam optimization in three different ways: (i) using the `higher` package (Grefenstette et al., 2019); (ii) PyTorch's autograd implementation; and (iii) our efficient reverse-mode implementation (Algorithm 1). We measure their effect on downstream performance as well as the time and memory associated with each outer-loop iteration in Figure 5. We first observe that Adam is much better suited for DD in our setting. This is a novel finding in the context of meta-learning and its applications, where previously Adam has been reported to be worse than SGD (Grefenstette et al., 2019). Further, we observe that while different reverse-mode implementations of Adam lead to data of similar sample quality, their computational properties vastly differ. We observe that PyTorch and `higher` have similar memory footprints, but the former has a lower runtime. Our efficient implementation elegantly trades-off memory with runtime, leading to constant memory footprint and a linear increase in runtime compared to PyTorch's autograd. This allows FARZI to scale to large autoregressive datasets without compromising on data fidelity.

Table 1: Cross-architecture generalization for FARZI DATA of size [50×150] of the ML-100k dataset. Note that the `student network` is used *only for training* on FARZI DATA, while the `teacher network` is used in FARZI's inner-loop. Further details about the following sequential recommendation models can be found in Appendix C.4.

| | Student HR@10 / HR@100 | | |
|---|---|---|---|
| Teacher | SASRec (Kang & McAuley, 2018) | GRU4Rec (Hidasi et al., 2016) | FMLP (Zhou et al., 2022a) |
| SASRec | 19.61/61.50 | 19.93/64.47 | 17.81/58.21 |
| GRU4Rec | 18.23/61.08 | 22.16/66.70 | 20.14/60.23 |
| Full-Data | 18.23/60.65 | 21.20/64.05 | 17.28/59.27 |

Figure 5: Changes in distillation performance and computational scalability of each outer-loop step for different inner-loop optimizers and increasing number of inner-loop steps. All results are for [10×150] sized FARZI DATA of the ML-100k dataset.

**How do different meta-objectives affect FARZI?** We further evaluate the importance of FARZI's optimization objective by comparing it with existing DD approaches. We adapt existing approaches to work with autoregressive data by reusing the latent distillation proposition of FARZI, and vary only the outer-loop *goodness function* to (i) gradient matching (DC (Zhao et al., 2021)); (ii) meta-matching (MM (Wang et al., 2018; Deng & Russakovsky, 2022)); or (iii) trajectory matching (MTT (Cazenavette et al., 2022)). See the formal definitions for each of these objectives in Appendix C.5. Even though all existing DD approaches use SGD in their inner-loop, we nonetheless experiment with both SGD and our efficient reverse-mode Adam (Algorithm 1), and list the results in Table 2. We observe that Adam is a consistently better inner-loop optimizer irrespective of the meta-objective used. This is in stark contrast with existing DD studies which use SGD in the inner-loop. Further, FARZI significantly outperforms all existing DD techniques despite improving them to use Adam in the inner-loop.

**How important are pre-trained trajectories for data distillation?** To elicit the importance of the pre-trained trajectories, *i.e.*, $\Omega \triangleq \{[\theta_i]_{i=1}^T \mid \theta_0 \sim \Theta\}$ in FARZI's optimization (Equation (2)), we plot the change in downstream distillation performance with increasing $|\Omega|$ in Figure 6b. We indeed observe a massive improvement in downstream distillation performance with using as little as just 5 trajectories, compared to randomly initializing networks in FARZI's inner-loop. Notably, the improvement saturates as we keep adding more trajectories to $\Omega$.

**Does FARZI affect cold users or items more?** A longstanding problem in recommender systems is modeling the cold-start scenario, *i.e.*, users/items with less data. We study the effect training models on FARZI DATA from the cold-start perspective, by stratifying the users and items based on their popularity into equal-sized quantiles, and checking the trained model's performance on each individual quantile. In Figure 6a, we do this for SASRec (Kang & McAuley, 2018) trained on (i) the full dataset; and (ii) FARZI DATA synthesized using different hyper-parameter combinations. We first observe that less popular items are harder to model, as is the typical case of recommender systems. Further, we observe that models trained on FARZI DATA are, in expectation, (i) better on the tail/torso

Table 2: Comparison of FARZI with other existing DD techniques modified to distill autoregressive data. Results for both using SGD or Adam as the inner-loop optimizer are listed. Meta-matching is shortened as MM. The best distillation result for each metric is colored **orange**, and the best result other than FARZI is colored **blue** for Adam-based methods and **emboldened** for SGD-based methods.

| Dataset | Metric | Random Sampling | Data Distillation Objectives | | | | | | FARZI | Full-Data |
|---|---|---|---|---|---|---|---|---|---|---|
| | | | DC | | MM | | MTT | | | |
| | | | SGD | Adam | SGD | Adam | SGD | Adam | | |
| ML-100k [50×150] | HR@10 ↑ | 7.74 | 7.95 | 11.77 | 9.65 | **16.86** | **12.19** | 14.52 | **19.61** | 18.23 |
| | HR@100 ↑ | 39.13 | 41.88 | 49.52 | 42.31 | **58.43** | **50.37** | 56.94 | **61.50** | 60.65 |
| | nDCG@10 ↑ | 3.83 | 3.44 | 5.6 | 4.72 | **8.35** | **6.12** | 6.73 | **9.91** | 9.33 |
| | nDCG@100 ↑ | 9.61 | 9.84 | 12.51 | 10.85 | **16.47** | **13.03** | 14.66 | **17.91** | 17.69 |
| PTB [400×50] | Perplexity ↓ | 218.66 | 203.23 | 131.07 | **180.61** | **115.84** | 202.98 | 129.72 | **91.92** | 72.10 |
| | Accuracy ↑ | 20.42 | 20.64 | 22.35 | **21.60** | **23.47** | 21.00 | 23.00 | **25.16** | 26.03 |

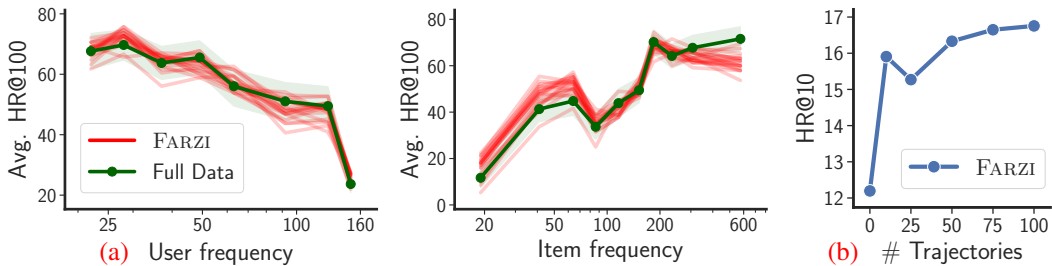

(a) User frequency          Item frequency          (b) # Trajectories

Figure 6: (a) Performance of SASRec trained on [50×150] sized FARZI DATA for ML-100k, and stratified over the popularity of users and items. The popularities are quantized into 10 equal sized bins and the average HR@100 is plotted. For results on more metrics see Appendix D, Figure 7. (b) Performance change of SASRec trained on [10×150] sized FARZI DATA for ML-100k with increasing number of pretrained trajectories. For results on more metrics see Appendix D, Figure 8.

region of users/items; but (ii) worse for the head users/items. Notably, this behaviour is not directly optimized-for by FARZI, and is a by-product of the overall data-quality optimization in Equation (2).

## 5 CONCLUSION & FUTURE WORK

In this paper, we proposed FARZI — a scalable technique to summarize large amounts of autoregressive data into a terse, high-fidelity data summary. Through extensive experiments on next-item recommendation and language modeling, we demonstrated that data synthesized by FARZI (FARZI DATA) is able to train various kinds of state-of-the-art models to the same quality (if not better) as training them on the full dataset, despite FARZI DATA being up to 3 orders of magnitude smaller.

Having demonstrated FARZI DATA's prowess to train autoregressive models, we also highlight a few shortcomings and unexplored directions that we delay for future work. First, even though FARZI performs distillation in a latent-space, it is hard to scale to applications that naturally consist of very-long sequences, *e.g.*, video, music, *etc.* because FARZI DATA is parameterized linearly in the length of each sequence. Further, scaling to larger models (*e.g.*, T5 (Raffel et al., 2020)) as well as larger datasets (*e.g.*, C4 (Raffel et al., 2019)) isn't as trivial due to practical constraints related to optimization and computational resources, but very important from a practical standpoint for future research, such as enabling cost-effective training of these large models on compact synthetic data.

Laying down the foundation for data distillation in autoregressive modeling, FARZI also opens up new research directions from varied angles. First, the ability to train higher quality models using less data is counter-intuitive and under-explored but also highly important from economical and environmental standpoints. Further, training models on differentialy private data summaries (Dong et al., 2022) instead of PII data can provide an added protection layer and be beneficial from copyright-protection, ethics, and fairness perspectives.

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

# A  ALGORITHMS

---

**Algorithm 2** Adam optimization (Kingma & Ba, 2015)

---

1: **Input:** initial $\mathbf{w}_0$, decays $(\beta_1, \beta_2)$, learning rate $\alpha$, constant $\epsilon$, loss function $L(\mathbf{w}, \mathbf{x})$
2: **Initialize:** $\mathbf{m}_0 \leftarrow 0$, $\mathbf{v}_0 \leftarrow 0$
3: **for** $t = 1$ to $T$ **do**
4:      $\mathbf{g}_t \triangleq \nabla_{\mathbf{w}} L(\mathbf{w}_{t-1}, \mathbf{x})$                                        ▷ evaluate gradient
5:      $\mathbf{m}_t = \beta_1 \cdot \mathbf{m}_{t-1} + (1 - \beta_1) \cdot \mathbf{g}_t$                       ▷ biased first moment estimate
6:      $\mathbf{v}_t = \beta_2 \cdot \mathbf{v}_{t-1} + (1 - \beta_2) \cdot \mathbf{g}_t^2$                     ▷ biased second moment estimate
7:      $\hat{\mathbf{m}}_t \triangleq \mathbf{m}_t / (1 - \beta_1^t)$                      ▷ bias-corrected first moment estimate
8:      $\hat{\mathbf{v}}_t \triangleq \mathbf{v}_t / (1 - \beta_2^t)$                    ▷ bias-corrected second moment estimate
9:      $\mathbf{w}_t = \mathbf{w}_{t-1} - \alpha \cdot \hat{\mathbf{m}}_t / (\hat{\mathbf{v}}_t + \epsilon)$                      ▷ update parameters
10: **Output:** trained parameters $\mathbf{w}_T$, biased first moment $\mathbf{m}_T$, biased second moment $\mathbf{v}_T$

---

# B  PROOFS

## B.1  PROOF OF THEOREM 3.1

*Proof.* We first begin by defining a few preliminary terms and properties:

**Definition B.1. (Representativeness of $\mathcal{D}$)** *For a given function-class $\mathcal{F}$, task loss function $l(\cdot)$, train-set $\mathcal{D} = \{x_1, x_2, \ldots, x_n\}$ sampled from the true data distribution $\mathcal{P}^n$:*

$$\mathrm{Rep}(\mathcal{F}, \mathcal{D}) \triangleq \sup_{f \in \mathcal{F}} \left( \mathop{\mathbb{E}}_{x \sim \mathcal{P}} [l(f, x)] - \mathop{\mathbb{E}}_{x \sim \mathcal{D}} [l(f, x)] \right) ,$$

*which intuitively measures the maximum discrepancy between the empirical risk and the true risk for a given training set. Naturally, a lower $\mathrm{Rep}(\mathcal{F}, \cdot)$ avoids overfitting and is desirable.*

**Definition B.2. (Rademacher complexity)** *For a given given function-class $\mathcal{F}$, and train-set $\mathcal{D} = \{x_1, x_2, \ldots, x_n\}$:*

$$\mathfrak{R}(\mathcal{F}, \mathcal{D}) \triangleq \mathop{\mathbb{E}}_{\sigma_1, \sigma_2, \ldots, \sigma_n \in \{\pm 1\}} \left[ \sup_{f \in \mathcal{F}} \frac{\sigma_i \cdot f(x_i)}{n} \right] ,$$

*where $\sigma_1, \sigma_2, \ldots, \sigma_n$ are independent random variables from the Rademacher distribution. $\mathfrak{R}(\mathcal{F}, \mathcal{D})$ intuitively measures the learning capacity of $\mathcal{F}$ by it's ability to fit random label assignments of $\mathcal{D}$.*

**Lemma B.3. (Lemma 26.2 in Shalev-Shwartz & Ben-David (2014, Chapter 26))**

$$\mathop{\mathbb{E}}_{\mathcal{D} \sim \mathcal{P}^n} [\mathrm{Rep}(\mathcal{F}, \mathcal{D})] \leq 2 \mathop{\mathbb{E}}_{\mathcal{D} \sim \mathcal{P}^n} [\mathfrak{R}(\mathcal{F}, \mathcal{D})]$$

**Lemma B.4. (Theorem 1 in Latorre et al. (2021))** *Let $\mathcal{F}_\lambda$ be the set of norm-bounded quadratic classifiers:*

$$\mathcal{F}_\lambda \triangleq \{f_{\mathbf{w}} : f_{\mathbf{w}}(x) = x^T \mathbf{w} x , \|w\| < \lambda\}$$

*Then for a training-set $\mathbf{X} \in \mathbb{R}^{n \times d}$:*

$$\mathfrak{R}(\mathcal{F}, \mathbf{X}) \lesssim \lambda \sqrt{\frac{r(\mathbf{X}^T \mathbf{X}) \log d}{n}} \|\mathbf{X}^T \mathbf{X}\|_2 \quad s.t. \quad r(\mathbf{X}^T \mathbf{X}) \triangleq \frac{\mathrm{trace}(\mathbf{X}^T \mathbf{X})}{\|\mathbf{X}^T \mathbf{X}\|_2} ,$$

*where, $r(\cdot)$ represents the intrinsic dimension of a PSD matrix (Tropp et al., 2015, Chapter 7).*

Now we're ready to prove Theorem 3.1. In our case, given the norm-bounded quadratic function-class $\mathcal{F}_\lambda$ and $\mathcal{D}_{\mathsf{naive}} \in \mathbb{R}^{\mu \times \xi \times \dim(\mathcal{V})}$ such that w.l.o.g $\|\mathcal{D}_{\mathsf{naive}}\|_2 = 1$:

$$\mathop{\mathbb{E}}_{\mathcal{D}_{\mathsf{naive}} \sim \mathbb{R}^{\mu \times \xi \times \dim(\mathcal{V})}} \left[ \mathrm{Rep}(\mathcal{F}_{\lambda}, \mathcal{D}_{\mathsf{naive}}) \right] \leq 2 \mathop{\mathbb{E}}_{\mathcal{D}_{\mathsf{naive}} \sim \mathbb{R}^{\mu \times \xi \times \dim(\mathcal{V})}} \left[ \mathfrak{R}(\mathcal{F}, \mathcal{D}_{\mathsf{naive}}) \right] \qquad \text{(Lemma B.3)}$$

$$\lesssim \mathop{\mathbb{E}}_{\mathcal{D}_{\mathsf{naive}} \sim \mathbb{R}^{\mu \times \xi \times \dim(\mathcal{V})}} \left[ 2\lambda \sqrt{\frac{r(\mathcal{D}_{\mathsf{naive}}^T \mathcal{D}_{\mathsf{naive}}) \log(\xi \dim(\mathcal{V}))}{\mu}} \right]$$

$$\text{(Lemma B.4)}$$

Furthermore, the intrinsic dimension of a PSD matrix $A$ obeys:

$$\begin{aligned}
r(A) &\triangleq \frac{\mathrm{trace}(A)}{\|A\|_2} = \frac{\sum_i \lambda_i(A)}{\lambda_{\mathsf{max}}(A)} \\
&\leq \frac{\lambda_{\mathsf{max}}(A) \cdot \mathrm{rank}(A)}{\lambda_{\mathsf{max}}(A)} \\
&\leq \mathrm{rank}(A)
\end{aligned}$$

where, the first line uses the alternate trace definition, and norm-eigenvalue equivalence. Combining the two findings:

$$\mathop{\mathbb{E}}_{\mathcal{D}_{\mathsf{naive}} \sim \mathbb{R}^{\mu \times \xi \times \dim(\mathcal{V})}} \left[ \mathrm{Rep}(\mathcal{F}_{\lambda}, \mathcal{D}_{\mathsf{naive}}) \right] \lesssim \mathop{\mathbb{E}}_{\mathcal{D}_{\mathsf{naive}} \sim \mathbb{R}^{\mu \times \xi \times \dim(\mathcal{V})}} \left[ 2\lambda \sqrt{\frac{\log(\xi \dim(\mathcal{V}))}{\mu}} \sqrt{\mathrm{rank}(\mathcal{D}_{\mathsf{naive}}^T \mathcal{D}_{\mathsf{naive}})} \right]$$

$$\lesssim 2\lambda \sqrt{\frac{\log(\xi \dim(\mathcal{V}))}{\mu}} \mathop{\mathbb{E}}_{\mathcal{D}_{\mathsf{naive}} \sim \mathbb{R}^{\mu \times \xi \times \dim(\mathcal{V})}} \left[ \sqrt{\mathrm{rank}(\mathcal{D}_{\mathsf{naive}})} \right]$$

$$\lesssim 2\lambda \sqrt{\frac{\log(\xi \dim(\mathcal{V}))}{\mu}} \cdot \min(\sqrt{\mu}, \sqrt{\xi \cdot \dim(\mathcal{V})}) \qquad (3)$$

On the other hand, for the non-parameterized formulation of FARZI DATA:

$$\mathop{\mathbb{E}}_{\substack{\tilde{\mathcal{D}}_{\mathsf{syn}} \sim \mathbb{R}^{\mu \times \xi \times d} \\ \mathbf{M} \sim \mathbb{R}^{d \times \dim(\mathcal{V})}}} \left[ \mathrm{Rep}(\mathcal{F}_{\lambda}, \tilde{\mathcal{D}}_{\mathsf{syn}} \cdot \mathbf{M}) \right] \lesssim 2\lambda \sqrt{\frac{\log(\xi \dim(\mathcal{V}))}{\mu}} \mathop{\mathbb{E}}_{\substack{\tilde{\mathcal{D}}_{\mathsf{syn}} \sim \mathbb{R}^{\mu \times \xi \times d} \\ \mathbf{M} \sim \mathbb{R}^{d \times \dim(\mathcal{V})}}} \left[ \sqrt{\mathrm{rank}(\tilde{\mathcal{D}}_{\mathsf{syn}} \cdot \mathbf{M})} \right]$$

$$\lesssim 2\lambda \sqrt{\frac{\log(\xi \dim(\mathcal{V}))}{\mu}} \cdot \sqrt{d} \qquad (4)$$

Finally, comparing Equations (3) and (4):

$$\mathop{\mathbb{E}}_{\tilde{\mathcal{D}}_{\mathsf{syn}}, \mathbf{M}} \left[ \mathrm{Rep}(\mathcal{F}, \tilde{\mathcal{D}}_{\mathsf{syn}} \cdot \mathbf{M}) \right] < \mathop{\mathbb{E}}_{\mathcal{D}_{\mathsf{naive}}} \left[ \mathrm{Rep}(\mathcal{F}, \mathcal{D}_{\mathsf{naive}}) \right] \quad \text{if} \quad d < \min(\mu, \xi \cdot \dim(\mathcal{V}))$$

$$\square$$

### B.2  PROOF OF PROPOSITION 3.2

*Proof.* Using the chain rule of derivatives:

$$\begin{aligned}
\frac{df(\mathbf{w}_T)}{d\mathbf{m}_0} &\triangleq d\mathbf{m} = d\mathbf{m} + \frac{\partial w_t}{\partial m_t} \cdot d\mathbf{w} \\
&= d\mathbf{m} - \frac{\alpha}{1 - \beta_1^t} \left( \frac{\left[ \sqrt{\hat{\mathbf{v}}_t} + \epsilon \right] - \left[ \mathbf{m}_t \cdot \left( \frac{\partial \mathbf{v}_t / \partial \mathbf{m}_t}{2 \cdot \sqrt{\hat{\mathbf{v}}_t} \cdot (1 - \beta_2^t)} \right) \right]}{(\sqrt{\hat{\mathbf{v}}_t} + \epsilon)^2} \right) \cdot d\mathbf{w} \\
&= d\mathbf{m} + \alpha' \cdot \left( \frac{\left[ \mathbf{m}_t \cdot \left( \frac{\partial \mathbf{v}_t / \partial \mathbf{m}_t}{2 \cdot \sqrt{\mathbf{v}_t}} \right) \right]}{(\sqrt{\mathbf{v}_t} + \epsilon')^2} - \frac{1}{\sqrt{\mathbf{v}_t} + \epsilon'} \right) \cdot d\mathbf{w} \quad ,
\end{aligned}$$

where, $\epsilon' \triangleq \epsilon \cdot \sqrt{1 - \beta_2^t}$ and $\alpha' \triangleq \frac{\alpha \cdot \sqrt{1 - \beta_2^t}}{1 - \beta_1^t}$.

Using the chain rule again:

$$\frac{\partial \mathbf{v}_t}{\partial \mathbf{m}_t} = \frac{\partial \mathbf{v}_t / \partial \mathbf{g}_t}{\partial \mathbf{m}_t / \partial \mathbf{g}_t} = \frac{2 \cdot (1 - \beta_2) \cdot \mathbf{g}_t}{(1 - \beta_1)} = 2 \cdot \beta' \cdot \mathbf{g}_t \quad,$$

where, $\beta' \triangleq \frac{1 - \beta_2}{1 - \beta_1}$, leading to finally:

$$\frac{df(\mathbf{w}_T)}{d\mathbf{m}_0} \triangleq d\mathbf{m} = d\mathbf{m} + \alpha' \cdot \left( \frac{\left[ \mathbf{m}_t \cdot \left( \frac{\partial \mathbf{v}_t / \partial \mathbf{m}_t}{2 \cdot \sqrt{\mathbf{v}_t}} \right) \right]}{(\sqrt{\mathbf{v}_t} + \epsilon')^2} - \frac{1}{\sqrt{\mathbf{v}_t} + \epsilon'} \right) \cdot d\mathbf{w}$$

$$= d\mathbf{m} + \alpha' \cdot \left( \frac{\beta' \cdot \mathbf{m}_t \cdot \mathbf{g}_t}{\sqrt{\mathbf{v}_t} \cdot (\sqrt{\mathbf{v}_t} + \epsilon')^2} - \frac{1}{\sqrt{\mathbf{v}_t} + \epsilon'} \right) \cdot d\mathbf{w}$$

$\square$

## C EXPERIMENTAL DETAILS

### C.1 METRICS

We present a formal definition of all metrics used in this paper for both sequential recommendation and language modeling tasks.

**Sequential Recommendation.** We start by outlining some notation for defining the metrics. Let the set of users in the test-set be denoted by $\mathcal{U}$ and the set of all items be denoted by $\mathcal{I}$. For each user $u \in \mathcal{U}$, we denote its set of positive interactions $\mathcal{I}_u^+ \subseteq \mathcal{I}$, and similarly define its set of negative interactions $\mathcal{I}_u^- \triangleq \mathcal{I} \backslash \mathcal{I}_u^+$. We now define the metrics for evaluating the quality of a recommender system $\Phi : \mathcal{U} \mapsto \mathcal{I}^k$ which generates a set of $k$ item recommendations, as follows:

- **AUC:** Intuitively defined as a threshold independent classification performance measure, AUC can also be interpreted as the expected probability of a recommender system ranking a positive item over a negative item for any given user. More formally, let $\Phi$'s underlying relevance predictor be $\Phi_{\text{logit}} : \mathcal{U} \times \mathcal{I} \mapsto \mathbb{R}$, then the AUC for $\Phi$ is defined as:

$$\text{AUC}(\Phi) \triangleq \mathop{\mathbb{E}}_{u \sim \mathcal{U}} \left[ \mathop{\mathbb{E}}_{i^+ \sim \mathcal{I}_u^+} \left[ \mathop{\mathbb{E}}_{i^- \sim \mathcal{I}_u^-} \left[ \Phi_{\text{logit}}(u, i^+) > \Phi_{\text{logit}}(u, i^-) \right] \right] \right]$$

- **HitRate (HR@k):** Also termed as Recall@k; HR@k estimates how many positive items are predicted in $\Phi$'s top-k recommendation list. More formally, the HR@k for $\Phi$ is defined as:

$$\text{HR@k}(\Phi) \triangleq \mathop{\mathbb{E}}_{u \sim \mathcal{U}} \left[ \frac{|\Phi(u) \cap \mathcal{I}_u^+|}{|\mathcal{I}_u^+|} \right]$$

- **Normalized Discounted Cumulative Gain (nDCG@k):** Unlike HR@k which gives equal importance to all items in the recommendation list, the nDCG@k metric instead gives a higher importance to items predicted higher in the recommendation list and performs logarithmic discounting further down. More formally, let $\text{index}(i, \Phi(u))$ denote the index of item $i$ in the *sorted* recommendation list $\Phi(u)$, then the nDCG@k for $\Phi$ is defined as:

$$\text{nDCG@k}(\Phi) \triangleq \mathop{\mathbb{E}}_{u \sim \mathcal{U}} \left[ \frac{\text{DCG}_u(\Phi)}{\text{IDCG}_u} \right]$$

$$\text{DCG}_u(\Phi) \triangleq \sum_{i \in \mathcal{I}_u^+} \frac{i \in \Phi(u)}{\log_2 \big( \text{index}(i, \Phi(u)) + 1 \big)} \quad ; \quad \text{IDCG}_u \triangleq \sum_{i=1}^{|\mathcal{I}_u^+|} \frac{1}{\log_2(i + 1)}$$

**Language Modeling.** We first use Perplexity (PPL) to evaluate language modeling performance. Perplexity quantifies how uncertain the model is when trying to predict the next word in a sequence, given the previous words. Given a sentence $\vec{x_i}$, which is tokenized into a sequence of tokens $[w_1, w_2, \cdots, w_{|\vec{x_i}|}]$, the sentence PPL is defined as:

$$\log_2 (\text{PPL}_i) \triangleq -\frac{1}{|\vec{x_i}|} \sum_i^{|\vec{x_i}|} \log_2 P(w_i | w_1, w_2, \cdots, w_{i-1})$$

where $P$ is the probability assigned by the language model to $w_i$ given the context of the previous words. Then, given a corpus $\mathcal{C}$ containing $N$ sentences $\mathcal{C} = \{\vec{x_1}, \vec{x_2}, \cdots, \vec{x_N}\}$, the perplexity over $\mathcal{C}$ is defined as the average PPL over the sentence PPLs:

$$\text{PPL}_{\mathcal{C}} \triangleq \frac{1}{N} \cdot \sum_i^N \text{PPL}_i$$

To better evaluate the generation quality of a language model, we also evaluate the average top-1 predicted token accuracy after greedy decoding, similar to the HR@1 metric described earlier.

## C.2 DATASETS

We list the datasets used in this paper as well as brief data statistics in Table 3. We discuss other task-specific preprocessing and train/test splitting strategy below.

**Sequential Recommendation.** Owing to recent work (Sachdeva et al., 2022c), we follow the minimal amount of preprocessing by only removing the users with less than two total interactions. We simulate the train/test split from the strong-generalization school-of-thought (Liang et al., 2018), where we keep a completely disjoint set of $80/10/10\%$ train, validation, and test users split randomly. For each user in the validation/test-set, the chronologically last interacted item is used for computing ranking metrics, whereas all previous interactions are used as context for the model. Further, to simulate a realistic recommendation scenario, we compute all metrics on the full item-space without any down-sampling (Krichene & Rendle, 2020). The definition of all metrics used in this paper can be found in Appendix C.1.

**Language Modeling.** We employ the Penn Treebank (PTB) dataset, an established and openly accessible benchmark extensively utilized in natural language processing and language modeling tasks, as introduced by (Marcus et al., 1993). We use the train/validation/test split of the official release. The original PTB corpus consists of more than 4.5 million words of American English, featuring a word vocabulary of 9,999 words, including the <unk> token. In our experimentation, we opt to maintain a vocabulary comprising 2,000 words with the highest frequencies, while any out-of-vocabulary words are represented as <unk>.

## C.3 HYPER-PARAMETERS

For the sake of better reproducibility, we list all hyper-parameter combinations tried for our experiments in Tables 4 and 5.

## C.4 ADDITIONAL DETAILS

We provide brief descriptions about all kinds of model architectures used in this paper for different experiments:

- **Transformer (Vaswani et al., 2017).** A causal transformer architecture for language modeling. The hyper-parameters are listed in Table 5.
- **SASRec (Kang & McAuley, 2018).** A causal transformer architecture for sequential recommendation. The hyper-parameters are listed in Table 4.
- **GRU4Rec (Hidasi et al., 2016).** An GRU-based architecture for sequential recommendation, trained using the cross-entropy loss. We use a single, 16-dimensional hidden-layer for the GRU4Rec architecture which was ascertained by conducting a grid-search on the ML-100k's validation-set.

- **FMLP (Zhou et al., 2022a).** An all-MLP architecture which replaces the self-attention blocks in SASRec with filter-enhanced MLPs for sequential recommendation. We use a single, 256-dimensional block for the FMLP architecture which was ascertained by conducting a grid-search on the ML-100k's validation-set.

## C.5 ALTERNATIVE DATA DISTILLATION OBJECTIVES

We provide a brief description and formal optimization of other existing data distillation objectives used in Section 4.2. Note that we list the modified optimization objectives where we use FARZI's latent factorization, and use $\mathrm{Opt}$ to denote the underlying inner-loop optimizer (SGD or Adam).

- **DC (Zhao et al., 2021):** This data distillation objective performs one-step gradient matching using a distance function $\mathfrak{D} : \mathbb{R}^{|\Phi|} \times \mathbb{R}^{|\Phi|} \mapsto \mathbb{R}$:

$$\underset{\mathbf{M}, \tilde{\mathcal{D}}_{\text{syn}}}{\arg\min} \; \underset{\theta_0 \sim \Theta}{\mathbb{E}} \left[ \sum_{t=0}^{T} \mathfrak{D} \left( \nabla_\theta \mathcal{L}_{\mathcal{D}}(\theta_t), \nabla_\theta \mathcal{L}_{\mathcal{D}_{\text{syn}}}(\theta_t) \right) \right]$$

$$\text{s.t.} \quad \theta_{t+1} \leftarrow \mathrm{Opt} \left( \theta_t, \nabla_\theta \mathcal{L}_{\mathcal{D}_{\text{syn}}}(\theta_t) \right) \quad ; \quad \mathcal{D}_{\text{syn}} \leftarrow \mathrm{softmax} \left( \tilde{\mathcal{D}}_{\text{syn}} \cdot \mathbf{M} / \tau \right) .$$

- **MM (Wang et al., 2018; Deng & Russakovsky, 2022):** The meta-matching objective computes the meta-gradient by unrolling the inner-loop optimization starting from random networks:

$$\underset{\mathbf{M}, \tilde{\mathcal{D}}_{\text{syn}}}{\arg\min} \; \underset{\theta_0 \sim \Theta}{\mathbb{E}} \left[ \mathcal{L}_{\mathcal{D}}(\theta_T) \right]$$

$$\text{s.t.} \quad \theta_{t+1} \leftarrow \mathrm{Opt} \left( \theta_t, \nabla_\theta \mathcal{L}_{\mathcal{D}_{\text{syn}}}(\theta_t) \right) \quad ; \quad \mathcal{D}_{\text{syn}} \leftarrow \mathrm{softmax} \left( \tilde{\mathcal{D}}_{\text{syn}} \cdot \mathbf{M} / \tau \right) .$$

- **MTT (Cazenavette et al., 2022):** The trajectory matching objective computes the meta-gradient by matching the parameters of networks trained on the real data for $M$ optimization steps *vs.* models trained on the data summary for $N \ll M$ steps. Let $\{\theta_t^{\mathcal{D}}\}_{t=0}^{T}$ represent the training trajectory of training $\Phi_\theta$ on $\mathcal{D}$, and $\mathfrak{D} : \mathbb{R}^{|\Phi|} \times \mathbb{R}^{|\Phi|} \mapsto \mathbb{R}$ be a pertinent distance function:

$$\underset{\mathbf{M}, \tilde{\mathcal{D}}_{\text{syn}}}{\arg\min} \; \underset{\theta_0 \sim \Theta}{\mathbb{E}} \left[ \sum_{t=0}^{T-M} \frac{\mathfrak{D} \left( \theta_{t+M}^{\mathcal{D}}, \theta_{t+N}^{\mathcal{D}_{\text{syn}}} \right)}{\mathfrak{D} \left( \theta_{t+M}^{\mathcal{D}}, \theta_t^{\mathcal{D}} \right)} \right]$$

$$\text{s.t.} \quad \theta_{t+i+1}^{\mathcal{D}_{\text{syn}}} \leftarrow \mathrm{Opt} \left( \theta_{t+i}^{\mathcal{D}_{\text{syn}}}, \nabla_\theta \mathcal{L}_{\mathcal{D}_{\text{syn}}}(\theta_{t+i}^{\mathcal{D}_{\text{syn}}}) \right) \quad ; \quad \theta_{t+1}^{\mathcal{D}_{\text{syn}}} \leftarrow \mathrm{Opt} \left( \theta_t^{\mathcal{D}}, \nabla_\theta \mathcal{L}_{\mathcal{D}_{\text{syn}}}(\theta_t^{\mathcal{D}}) \right) \quad ;$$

$$\mathcal{D}_{\text{syn}} \leftarrow \mathrm{softmax} \left( \tilde{\mathcal{D}}_{\text{syn}} \cdot \mathbf{M} / \tau \right) .$$

# D ADDITIONAL RESULTS

We plot extended plots for the experiments conducted in Section 4.2:

- In Table 6, we plot the sample quality results of FARZI DATA in a tabular format for all datasets and metrics.
- In Figure 7, we analyze FARZI DATA's effect on cold users and cold items for all metrics described in Appendix C.1.
- In Figure 8, we analyze FARZI's reliance on the number of pretrained trajectories for all metrics described in Appendix C.1.
- In Figure 9, we plot the sample efficiency of FARZI DATA for sequential recommendation for all metrics described in Appendix C.1.

Table 3: Datasets used in this paper as well as a brief set of statistics.

| Dataset | # Users / # Sentences | # Items / # Unique tokens | # Interactions / # Total tokens | Seq. Length Mean / Median / Min |
|---|---|---|---|---|
| **Amazon Magazine** (Ni et al., 2019b) | 3k | 1.3k | 12k | 4.10 / 3 / 3 |
| **ML-100k** (Harper & Konstan, 2015) | 943 | 1.6k | 100k | 104.04 / 63 / 18 |
| **ML-1M** (Harper & Konstan, 2015) | 6k | 3.7k | 1M | 165.22 / 95 / 20 |
| **Netflix** (Bennett et al., 2007) | 476k | 17k | 100M | 210.91 / 98 / 3 |
| **PTB** (Marcus et al., 1993) | 49k | 10k | 1M | 22 / 21 / 2 |

Table 4: List of all hyper-parameters combinations tried for FARZI and other baselines for sequential recommendation.

| | Hyper-Parameter | Model | Magazine | ML-100k | ML-1M | Netflix |
|---|---|---|---|---|---|---|
| | Latent size | SASRec GRU4Rec FMLP | | {8, 16, 32, 50, 64, 128} | | |
| | # Layers | SASRec GRU4Rec FMLP | | {1, 2} | | |
| | Attention Heads | SASRec FMLP | | {1, 2} | | |
| | Learning rate | SASRec GRU4Rec FMLP | | {0.01, 0.02, 0.05} {0.01, 0.02, 0.05} {0.0001, 0.0002, 0.0005} | | |
| | Dropout | SASRec GRU4Rec FMLP FARZI | | {0.0, 0.2, 0.4} | | |
| | $\xi$ | FARZI | {10, 20} | {50, 100, 150} | {50, 100, 150} | 200 |
| | $d$ | FARZI | 8 | 8 | 32 | 32 |
| | $\tau$ | FARZI | | {0.5, 1, 2} | | |
| | $|\Omega|$ | FARZI | 100 | 100 | 100 | 50 |
| Inner loop | Weight Decay Learning Rate # Steps $\beta_1$ $\beta_2$ SGD Momentum | FARZI | | {0, $10^{-6}$} {0.01, 0.02} {100, 200, 300} 0.9 0.999 {0.5, 0.75, 0.9, 0.95, 0.99} | | |
| Outer loop | Weight Decay Learning Rate # Steps | FARZI | | {0, $10^{-6}$, $10^{-4}$} 0.01 4000 | | |
| Batch size | $\mathcal{D}$ $\mathcal{D}_{\mathsf{syn}}$ | FARZI | — | 512 — | 50 | 25 |

Table 5: List of all hyper-parameters combinations tried for FARZI and other baselines for language modeling.

| Hyper-Parameter | | Model | Penn Treebank |
|---|---|---|---|
| Latent size | | Transformer RNN | 16 |
| # Layers | | Transformer RNN | 1 |
| Attention Heads | | Transformer RNN | 1 |
| Learning rate | | Transformer RNN | {0.01, 0.02, 0.05} {0.01, 0.02, 0.05} |
| Dropout | | Transfomer RNN | {0.0, 0.2} |
| $\xi$ | | FARZI | {5, 15} |
| $d$ | | FARZI | 8 |
| $\tau$ | | FARZI | {0.5, 1, 2} |
| $|\Omega|$ | | FARZI | 400 |
| Inner loop | Weight Decay Learning Rate # Steps $\beta_1$ SGD Momentum | FARZI | {0, $10^{-7}$} {0.01, 0.02} {200, 300, 400, 500, 600} 0.999 - |
| Outer loop | Weight Decay Learning Rate # Steps | FARZI | {0, $10^{-6}$, $10^{-4}$} 0.01 8000 |
| Batch size | $\mathcal{D}$ $\mathcal{D}_{\text{syn}}$ | FARZI | 256 — |

Table 6: Performance change of SASRec & Transformer with various sizes of FARZI DATA for sequential recommendation & language modeling tasks respectively. The best result for each dataset & metric is colored **orange**.

| Dataset & Model | FARZI DATA size | HR@10 | HR@100 | nDCG@10 | nDCG@100 | AUC | PPL | Acc. |
|---|---|---|---|---|---|---|---|---|
| Magazine & SASRec | [10 x 10] ≡ 0.3% | 23.3 | **52.3** | 15.8 | 21.1 | **0.8428** | - | - |
| | [25 x 20] ≡ 0.8% | 23.9 | **52.3** | 16.5 | 21.6 | 0.8307 | - | - |
| | [50 x 20] ≡ 1.6% | **24.5** | 52.1 | **17.1** | **22.1** | 0.8291 | - | - |
| | Full-data | 23.2 | 52.0 | 16.9 | 21.7 | 0.8223 | - | - |
| ML-100k & SASRec | [10 x 150] ≡ 1% | 17.3 | 61.2 | 9.2 | 17.7 | 0.8957 | - | - |
| | [25 x 150] ≡ 2.6% | 19.3 | 61.6 | 9.9 | 17.7 | 0.902 | - | - |
| | [50 x 50] ≡ 5.3% | **19.6** | **62.9** | 9.9 | **18.1** | **0.9016** | - | - |
| | [100 x 100] ≡ 10.6% | 19.5 | 61.9 | **10.1** | **18.1** | **0.9016** | - | - |
| | Full-data | 18.2 | 60.6 | 9.3 | 17.6 | 0.9011 | - | - |
| ML-1M & SASRec | [10 x 150] ≡ 0.1% | 22.4 | 59.0 | 12.0 | 19.0 | 0.923 | - | - |
| | [50 x 100] ≡ 0.8% | 24.8 | 61.6 | 13.8 | 20.8 | 0.9301 | - | - |
| | [100 x 100] ≡ 1.6% | 25.6 | **63.6** | 14.1 | 21.3 | **0.9317** | - | - |
| | [200 x 50] ≡ 3.3% | 25.4 | 61.8 | 14.1 | 21.0 | 0.9315 | - | - |
| | [500 x 50] ≡ 8.2% | **26.2** | 61.0 | 13.8 | 20.7 | 0.9293 | - | - |
| | Full-data | **26.2** | 62.8 | **14.4** | **21.8** | 0.9291 | - | - |
| Netflix & SASRec | [50 x 200] ≡ 0.01% | 15.6 | 38.0 | 9.9 | 14.1 | 0.9235 | - | - |
| | [500 x 200] ≡ 0.1% | 17.8 | 40.7 | 11.6 | 16.1 | 0.9449 | - | - |
| | [2000 x 200] ≡ 0.4% | 17.5 | 40.3 | 11.3 | 15.8 | 0.9455 | - | - |
| | Full-data | **18.1** | **41.9** | **11.8** | **16.4** | **0.947** | - | - |
| PTB & Transformer | [10 x 50] ≡ 0.02% | - | - | - | - | - | 238.5 | 20.48 |
| | [200 x 50] ≡ 0.47% | - | - | - | - | - | 124.0 | 24.0 |
| | [400 x 50] ≡ 1% | - | - | - | - | - | 91.9 | 25.16 |
| | [2000 x 50] ≡ 4.7% | - | - | - | - | - | 91.0 | 25.4 |
| | Full-data | - | - | - | - | - | **72.10** | **26.03** |

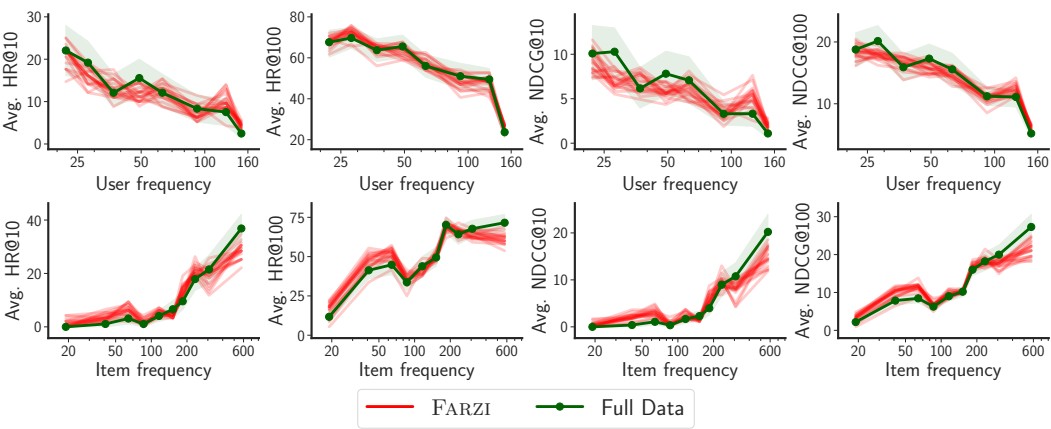

Figure 7: Performance of SASRec trained on [50×150] sized FARZI DATA of the ML-100k dataset, and stratified over the popularity of users and items. The user/item popularity spectrum is quantized into 10 equal sized bins and the average corresponding metric is plotted.

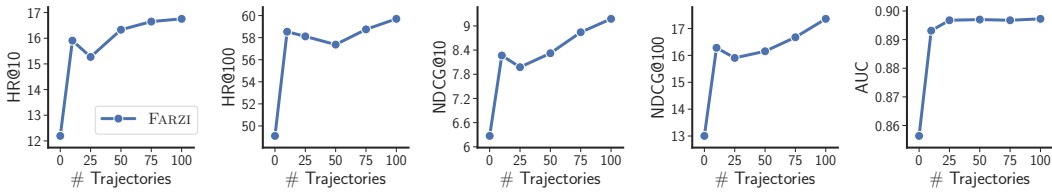

Figure 8: Performance change of SASRec trained on `[10×150]` sized FARZI DATA of the ML-100k dataset with increasing number of pretrained trajectories.

Figure 9: Performance change of SASRec model with increasing data summary size (log-scale) for sequential recommendation.

