# OpenReview forum: "Farzi Data: Autoregressive Data Distillation"
_ICLR.cc/2024/Conference — Submitted to ICLR 2024_

### Official Review · Reviewer_LDAP · 2023-10-17

**Soundness:** 2 fair
**Presentation:** 2 fair
**Contribution:** 3 good
**Rating:** 3
**Confidence:** 4

**Summary:**

The authors introduce a dataset distillation (DD) method called Farzi Data for data with a "left to right" (autoregressive) causal structure. Their algorithm has two novel elements: 1) the parameterization of the synthetic distilled data, which allows them to apply it to discrete data (such as the tokens in language modeling); and 2) a method for computing the outer loop gradient for DD when the inner loop is performed with Adam, which has a constant memory footprint independent of the number of inner optimization steps. They conduct extensive experiments with their proposed method on language modeling and sequential recommendation tasks. Compared to existing DD methods (adapted to discrete data via their parameterization), they obtain improved performance across the tested datasets, often obtaining downstream performance better than training a model on the entire original dataset.

**Strengths:**

**Algorithmic Contribution.** Algorithm 1 for computing the gradient through the inner-loop optimization with Adam using constant memory is a significant contribution. Among existing dataset distillation methods, those which take into account the entire training trajectory on the distilled data tend to obtain better accuracy (as compared to other methods which use surrogates for this objective such as the gradient matching objective in dataset condensation). However, the computational burden of these methods (specifically the memory requirement, which necessitates keeping the entire computation graph) renders them infeasible for application to larger datasets. Farzi Data takes a significant step towards addressing this problem by introducing an algorithm for differentiating through an inner loop optimized with Adam, whose memory does not scale with the number of steps in the inner loop (see Fig. 5). This is an important improvement for DD to be practically useful in real ML applications.

**Empirical Results.** The empirical results are also impressive. The authors obtain better performance than competing methods across several different real-world benchmarks. There are even scenarios where their distilled data consistently outperforms training on the entire original dataset (cf. Table 1), indicating that Farzi Data implicitly promotes some sort of "data cleaning" whereby samples that *hurt* model performance are removed or discounted. This is similar to, e.g., removing mislabeled points or data with negative Shapley values, but Farzi Data is not explicitly trained for this task.

**Weaknesses:**

**Presentation and Clarity.** While the actual prose of the paper was generally clear and easy to read, there are some major concerns with notation/presentation that limit understanding of some of the main contributions of the paper.

P1. There are many cases where important notation is not defined. For instance, $\mathrm{Rep}(\mathcal{F}, \mathcal{D})$ is defined in the Appendix, but not the main text, and is critical to interpreting Theorem 3.1. It is not stated what the terms $d\mathbf{m}$, $d\mathbf{x}$, and $d\mathbf{w}$ in Algorithm 1 are supposed to be, so it is impossible to determine if the expressions are correct or not. How to construct the output of the algorithm from these quantities is also not clear. What is the correspondence of the quantities in Alg. 1 to the DD problem, i.e., what will we actually update using the meta-gradient once we know how to compute it? Some (but not all) of these details can be found in the Appendix, but as they are critical to being able to understand the results, they should be moved to the main text and given appropriate explanations.

P2. Stylistically, there is also some nonstandard notation. For instance, $\mathcal{O}(100)$ (3rd bullet point, pg. 2). I suppose the authors meant "on the order of 100x", but big-O notation has a mathematically precise meaning that doesn't make sense here. Another instance is Proposition 3.2. "Correctness of Algorithm 1, Line 13" is not a complete mathematical statement (or a complete sentence). The result should be stated completely and precisely.

**Theoretical Results.** There are also issues with the theoretical results.

T1. The most critical problem is that the proof of the main theorem (Theorem 3.1) is not mathematically sound. Specifically, the authors want to show that the expected representativeness of their low-rank synthetic data parameterization is strictly less than the expected representativeness of a naive synthetic data parameterization, under some suitable conditions and for quadratic classifiers: $\mathbb{E}[\mathrm{Rep}(\mathcal{F}, \mathcal{D}_F)] < \mathbb{E}[\mathrm{Rep}(\mathcal{F}, \mathcal{D}_N)]$. ($\mathcal{D}_F$ and $\mathcal{D}_N$ stand for Farzi and naive data, respectively.) In their proof in Appendix B.1, they show that $\mathbb{E}[\mathrm{Rep}(\mathcal{F}, \mathcal{D}_F)] < B_1$ and $\mathbb{E}[\mathrm{Rep}(\mathcal{F}, \mathcal{D}_N)]$ for some bounds $B_1$ and $B_2$. Then, since $B_1 < B_2$, they conclude the desired result. This is not valid: $a < b$, $c < d$, and $b < d$ does not imply that $a < c$. There needs to be a _lower_ bound on the representativeness for the naive parameterization.

I remark that I believe the _result_ is (at least "morally") correct. The theorem essentially reduces to saying that the Rademacher complexity resulting from the low-rank parameterization is smaller than the Rademacher complexity from a general parameterization, which is intuitively obvious. However, the _proof_ has a fatal error and must be corrected somehow.

T2. For Lemma B.3 to hold, there must clearly be some assumptions on the loss function $l$; in order to apply the lemma from Shalev-Shwartz, the Rademacher complexity of the loss composed with the models in $\mathcal{F}$ must be considered, not $\mathcal{F}$ itself. As stated, I believe this lemma is not correct and the loss must be accounted for. Apart from the logical error, the motivation for the use of quadratic classifiers in the theorem wasn't clear to me. What connection do such models have to the auto-regressive tasks that Farzi Data is applied to?

T3. This is related to the presentation problems regarding the notation used in Algorithm 1, but the proof of Proposition 3.2 is also suspect. What is meant by $d\mathbf{m} = d\mathbf{m} + \frac{\partial w_t}{\partial m_t} \cdot d\mathbf{w}$? Is $w_t$ supposed to be $\mathbf{w}_T$, or is this expression meant to be a recursive formula? What about the formulas for the other quantities, and how are these combined to compute the meta gradient?

If these issues can be satisfactorily addressed, along with the questions in the section below, I would be willing to raise my score to accept, given how promising the empirical results are.

**Questions:**

Q1. The authors mention that training with the reference trajectories $\Omega$ is important for obtaining the best performance, as compared with training only from randomly initialized networks. However, it wasn't clear to me if this might just have been the result of a greater number of training steps when learning the distilled dataset. That is, are the results in Fig. 6(b) with the total number of meta-gradient steps constant, or do the additional precomputed trajectories result in more meta-gradient steps?

Q2. On a related note, it was not clear to me exactly how the precomputed trajectories were used. My assumption was that instead of training the network in the inner loop only from random initializations, instead the network from the inner loop will be initialized with parameters from one of the training trajectories. Is this correct?

Q3. Why isn't FMLP also used as a teacher network in Table 1?

---

> ### Author Response · Authors · 2023-11-23
> **Response to reviewer LDAP (Part 1/2)**
>
> First of all, we really appreciate your review — we believe this is a good quality review where you ask very valid questions; irrespective of your low score recommendation. However, we do hope our responses shed some light on your key confusions and convince you to improve your score. It would really help us push this important line of work to one of the most technically capable audiences, at ICLR. Please let us know if we can add / expand on something:
>
> **Q: Important notation is not defined in the main-text.**
>
> A: Thank you for raising these super-valid points! We will make sure to address the following changes as you suggested to improve the readability of our paper:
> - **what the terms $d\mathbf{m}, d\mathbf{x}, d\mathbf{w}$**: We apologize for missing this detail. $d\mathbf{m}, d\mathbf{x}, d\mathbf{w}$ are the meta-gradients of $f(w_T)$ w.r.t $m_0, x, w_0$ respectively. Most of this notation is re-used from [1], but we will make sure to add this to the main-text.
> - **what will we actually update using the meta-gradient in Algorithm 1**: Using simple, gradient based optimization in the outer loop (we use Adam), once we have the meta-gradient from Algorithm 1, i.e., $d\mathbf{x} \triangleq df(w_T) / d\mathbf{x}$; we use this gradient to update $\mathbf{x} \leftarrow Adam(\mathbf{x}, d\mathbf{x})$ in the standard way. Note that $\mathbf{x}$ in our case is parameterized as $\tilde{\mathcal{D}}_{syn}$ and $\mathbf{M}$.
> - **non-standard $\mathcal{O}(100)$**: Absolutely, we’ll make sure to write 100x instead of $\mathcal{O}(100)$. Thanks!
> - **Statement of Proposition 3.2**: We’ll make sure to write the complete statement in the main-text instead of the current way. Thank you for pointing this out!
>
> **Q: Issue in Theorem 3.1**
>
> A: Thank you for catching our honest mistake at the end of Theorem 3.1 — we are highly appreciative! We tried taking the lower-bound approach as you suggested but to no-end: there are no commonly available lower-bounds for Radamacher complexities because of the underlying $\sup()$ in its definition. Even trying to develop novel lower bounds isn’t a trivial process and/or contribution. Hence, given the short timeline of this rebuttal, we have decided to remove Theorem 3.1 from our paper and totally admit to our mistake.
>
> We do hope that you believe that removing this Theorem doesn’t hurt our technical and practical contributions too much, as you have yourself suggested in your review.
>
> **Q: Notation questions in Proposition 3.2**
>
> A: Thanks for pointing this out - we can definitely improve upon the clarity here. You’re correct by mentioning that this is a recursive formula, starting from $t=T$ going all the way to $t=1$. As outlined in Algorithm 1, $d\mathbf{m}, d\mathbf{x}, d\mathbf{w}$---which are the meta-gradients of $f(w_T)$ w.r.t $m_0, x, w_0$ respectively---are variables which are carried over, with the equations in Algorithm 1 defining the update rules for our recursive meta-gradient calculation. Please note that $d\mathbf{x}$ which represents $df(w_T) / d\mathbf{x}$ is the final meta-gradient used for updating our synthetic data ($\tilde{\mathcal{D}}_{syn}$ and $\mathbf{M}$) that we’re interested in. Similarly, for other meta-learning applications like learning suitable weight initializations (e.g., MAML), we can use the same Algorithm 1 to estimate other meta-gradients like $d\mathbf{w}$ which represents $df(w_T) / dw_0$, if we use Adam in the inner-loop.
>
> Please note that our Algorithm 1 shares the same spirit as the popular reverse-mode SGD algorithm with its recursive formulation [1], albeit much simpler due to being a first-order optimizer.
>
> **Q: Are the results in Fig. 6(b) with the total number of meta-gradient steps constant?**
>
> A: Great question again! We would like to mention here that all of the results in Fig. 6(b) are performed with the outer loop running till convergence of a maximum of 4000 steps. Please note that realistically, convergence for all settings in our paper happens between 1000-2000 steps, at most. We’ll make sure to add a clarification in the main text.
>
> We would also like to provide some added intuition behind using the pre-trained trajectories as is currently used in Farzi. In the model weight space, $\Omega$ contains a set of points with varying levels of quality (models are trained starting from random till convergence, with intermediate checkpoints being stored). Farzi’s inner loop, using these points as starting points, now has a better exploration of the entire model weight space, and the final data will be better optimized to train models of varying quality. This makes the final data summary more robust and enables it to train models at different stages of training.
>
> [1] Maclaurin, Dougal, David Duvenaud, and Ryan Adams. "Gradient-based hyperparameter optimization through reversible learning." International conference on machine learning. PMLR, 2015.

---

> > ### Author Response · Authors · 2023-11-23
> > **Response to reviewer LDAP (Part 2/2)**
> >
> > **Q: How exactly were the precomputed trajectories used?**
> >
> > A: Thanks for pointing this out! There is definitely a clarification to be made here: $\Omega$ contains the *flat* set of all episodic checkpoints for the training trajectories and when we sample $\theta_0$ (0 here represents the inner loop time-index), it can come from the union of all the intermediate checkpoints available to us ($\Omega$). We’ll make sure to add the clarification in the pdf as well.
> >
> > **Q: Why isn't FMLP also used as a teacher network?**
> >
> > A: Great catch! This is motivated primarily by FMLP being very similar to SASRec in its architecture and inductive biases and we’re mostly interested in ascertaining cross-architecture generalization, e.g., Transformer $\mapsto$ RNN and vice-versa. This allows us to save experiment time and not compute the full 3x3 matrix in Table 1. We’ll make sure to add this clarification in the main text.
> >
> > We would like to end by once again requesting you to kindly reconsider your final rating as it strictly decides the outcome of this paper, which in your own self-opinion contains significant technical contributions along with strong empirical evidence.

---

> ### Comment · Reviewer_LDAP · 2023-11-28
>
> Thanks to the authors for their detailed response. I believe your proposed changes will improve the paper. However, as it stands, I still cannot recommend acceptance for the following reasons:
>
> 1. I don't see an updated pdf, so I cannot verify whether or not the updates to the notation, definitions, etc. were sufficient for the paper to be acceptable for publication.
> 2. Regarding Theorem 3.1, I agree with the authors that simply removing it from the paper would be acceptable. However, the result of this theorem was used as motivation for the novelty of your procedure when responding to other reviewer comments (last question in the response to Reviewer THWQ), so there is still a problem here as well. It also forms a significant part of the paper as presented, and if it is removed, this may have a large impact on other reviewers' scores.
>
> In short, I don't believe it's possible for the proposed changes to be fairly assessed for the paper to be accepted to ICLR. I recommend the authors to revise the paper and resubmit to a future conference.

---

### Official Review · Reviewer_THWQ · 2023-10-31

**Soundness:** 2 fair
**Presentation:** 2 fair
**Contribution:** 3 good
**Rating:** 5
**Confidence:** 2

**Summary:**

The paper provides an extension of dataset distillation to sequence modeling along with a few other innovations, such as a low rank approximation of the distilled dataset and an efficient trick to save memory during meta-learning. Overall, the paper contains strong (albeit limited) empirical results on the sequence modeling (penn tree bank) and recommendation systems datasets.

**Strengths:**

* The high level motivation of the problem is quite the need of the hour, as with larger models we need to better understand their dependencies on the data
* Pursuit of this research direction could potentially yield methods that enable us to train SOTA transformer models for a fraction of the input cost
* Empirical results are thorough, although a bit limited in terms of number of datasets for sequence modeling (only PTB is used)

**Weaknesses:**

A number of points about the approach were unclear to me from the writeup, and I would appreciate clarifications from the authors:

* It is said that the complexity of the dataset distillation algorithm scales by the size of the vocabulary (page. 4) and the size of the sequence that we wish to model. I can see the latter to be the case, since the loss will now be summed over the entire sequence as opposed to one forward pass (so the complexity of the forward pass is increased). However, I do not see how the time complexity increases with the vocabulary size. Do we mean space complexity? Also, more than the forward pass the dominant factor in dataset distillation is the computation of a bunch of hessian vector products in the meta gradient. Those terms do not depend on the vocabulary size either… please clarify..
* It would be nice to provide an intuition for what is saving the memory, making things O(1) in memory.  Currently the big algorithm block does not provide an intuition for how this approach is O(1) in memory regardless of the number of timesteps of unrolling. This is important to clarify, since this is an important contribution, if clearly explained. If this approach is essentially gradient checkpointing, then it is worth noting that Deng and Russakovsky already implement a version of this in their code.
* Looking at Eqn. 2, I am a bit puzzled as to how \Omega, namely the trajectories from the real data are incorporated in the DD process. From what I am able to understand, \theta_0 \sim Omega -- namely the init is sampled from the pretrained trajectories, and then from the right hand side of eqn. 2 I understand that the rest of the trajectory is obtained using Adam on the synthetic data. Where is the role of the pretrained trajectories then? Please explain..

* Rank regularization has been done in the previous work (Deng and Russakovsky) for dataset distillation. It should be cited that this has been done, and not be presented as a novelty..

**Questions:**

My major questions concern the clarifications about the approach listed above, without which it is really hard to judge the technical correctness / soundness of the paper.

---

> ### Author Response · Authors · 2023-11-23
> **Response to reviewer THWQ**
>
> We really thank you for your time and effort in reviewing the paper and providing us with your valuable feedback! Below are the responses to your well-motivated questions, which we hope will shed more clarity on Farzi’s design. Please let us know if we can add / expand on something:
>
> **Q: I do not see how the time complexity increases with the vocabulary size.**
>
> A: Looking at Farzi’s time complexity in Section 3, both the time and space complexities depend directly on the vocabulary size, i.e., $\dim(\mathcal{V})$. Let me provide further intuition on this:
> - Taking the naive, non-parameterized Farzi data parameterization as a simpler reference, each entry in the $\mu \times \xi \times \dim(\mathcal{V})$ data summary is a *parameter* in the outer-loop optimization. Hence, both the sequence length ($\xi$) and the vocabulary size ($\dim(\mathcal{V})$) directly contribute to the space complexity, as well as the time complexity, e.g., for computing and updating the meta-gradient for each of these parameters, that too for each of the $T$ steps in the inner loop.
>
> **Q: What is making things O(1) in memory? Is it essentially gradient checkpointing?**
>
> A: Thank you for the question! Definitely makes sense to provide an intuition for this. The key technological advancement here is Algorithm 1, and to be more specific Lines 13-15, which derive the reverse-mode meta-gradient calculation equations, when Adam is used in the inner-loop. Having access to these equations, we can now only keep the $d\mathbf{m}, d\mathbf{x}, d\mathbf{w}$ vectors as loop carry-over variables and compute the exact meta-gradient for updating the data summary (i.e., $d\mathbf{x}$) in $O(T)$ time using the for-loop in Algorithm 1, and making the space complexity independent of $T$. However, naive autodiff and standard meta-gradient libraries like higher would store *all intermediate variables of the inner-loop* making the space complexity linear in terms of $T$.
>
> Further, regarding the comparison to (Deng and Russakovsky), please note that they first of all leverage prior work [1] which developed reverse-mode SGD (similar to our reverse-mode Adam, albeit much simpler due to being a first-order optimizer). Next, reverse-mode Adam is completely different from gradient checkpointing in the case of meta-learning where it’s non-trivial and suboptimal [2]. On the other hand, our reverse-mode Adam is correct (optimal) because of our hand-derived meta-gradient calculations (Proposition 3.2).
>
> **Q: How is $\Omega$—the trajectories from the real data—incorporated in the DD process?**
>
> A: Thanks for pointing this out! There is definitely a clarification to be made here: $\Omega$ contains the *flat* set of all episodic checkpoints for the training trajectories and when we sample $\theta_0$ (0 here represents the inner loop time-index), it can come from the union of all the intermediate checkpoints available to us ($\Omega$). We’ll make sure to add the clarification in the pdf as well.
>
> **Q: Rank regularization has been done in previous work, and not be presented as a novelty.**
>
> A: Thanks for pointing this out! We do not intend to claim latent parameterization of data, in its entirety as our novelty. However, we sincerely believe the following pieces of development to be critical and also novel to our propositions:
> - Previous work regarding parameterized data distillation (not just (Deng and Russakovsky)) have been motivated to achieve better performance using fewer total bytes to store the data. However, in our setting of autoregressive data distillation, we would like to explicitly note that naive data parameterization is infeasible computationally due to the vocabulary size dimension in our data.
> - We provide formal connections as to why rank regularization *improves* performance by reducing overfitting and promoting implicit regularization. Please note that while this conclusion is intuitive, we are the first to showcase a formal proof.
>
> We would like to end by requesting you to kindly reconsider your final rating as it strictly decides the outcome of this paper, which in your own self-opinion is well-timed and contains strong empirical evidence.
>
> [1] Maclaurin, Dougal, David Duvenaud, and Ryan Adams. "Gradient-based hyperparameter optimization through reversible learning." International conference on machine learning. PMLR, 2015.
>
> [2] Hascoet, Laurent, and Mauricio Araya-Polo. "Enabling user-driven checkpointing strategies in reverse-mode automatic differentiation." arXiv preprint cs/0606042 (2006).

---

### Official Review · Reviewer_htDK · 2023-11-01

**Soundness:** 4 excellent
**Presentation:** 4 excellent
**Contribution:** 3 good
**Rating:** 6
**Confidence:** 3

**Summary:**

This paper proposes a method for distillation of "auto-regressive data", in this case meaning any data that is represented as event sequences. This can include natural language text, but also general time-series data. Their method aims to summarize a dataset into a sequence of latent embeddings (which can subsequently be decoded) given a downstream task such that they achieve similar performance to training on the complete dataset. They do this through a meta-learning procedure, optimizing directly through Adam for data which lowers downstream task loss.

**Strengths:**

My review comes from the point of view of someone familiar with training on natural language (and associated downstream evaluation), but not general event forecasting problems. I was not familiar with the benchmarks used by the author prior to reading this paper.

**Originality and Significance**

- The paper seems original. Aspects of this work (e.g. using meta-learning/second order methods) for distillation have been touched on in the past, but usually for smaller datasets, and generally not for auto-regressive tasks. Most past works I have seen which work on large corpuses revolve around finding mixing coefficients for existing datasets [1]. This method doesn't work on datasets of that size, however this shows an improvement in scaling.
- Getting a meta-learning approach to work on such dataset sizes is quite difficult, given difficulties with estimating second-order components over the full dataset. Scaling this to even larger language-style datasets would be an interesting (future) contribution.



**Quality and Clarity**

This paper is quite well-written. Experimental details are clear, and the method is properly motivated. Diagrams clarify the algorithm and the key difficulties to this method are highlighted appropriately.

[1] The Pile: An 800GB Dataset of Diverse Text for Language Modeling, Gao et al. 2021

**Weaknesses:**

**Weaknesses**

- The authors touch on language datasets as a motivation, however do not study this (or other large-sequence tasks) due to practical model/sequence length scaling constraints. Are there reasonable paths forward that would allow this to scale to longer sequence lengths/larger models?
- Given that the outer loop evaluates across the full original dataset, and the inner loop needs to be run several times to get updated parameters (Figure 5), what's the overall cost saving versus just training a model on the original dataset for more time (until matching student performance), if any?
- Have the authors thought about cases where there is significant noise in the training corpus? Given that the loss is computed with respect to the original dataset, it seems like this could be a problem if one ever tried to directly filter a noisy web-crawl.

**Questions:**

All questions have been included in the "Weaknesses" section above.

---

> ### Author Response · Authors · 2023-11-23
> **Response to reviewer htDK**
>
> We really thank you for your time and effort in reviewing the paper and providing us with your valuable feedback! Below are the responses to your well-motivated questions, which we hope will shed more clarity on Farzi’s design. Please let us know if we can add / expand on something:
>
> **Q: Are there reasonable paths forward that would allow this to scale to longer sequence lengths/larger models?**
>
> A: Thanks for the question! Regarding longer sequence lengths, even in most LLM pre-training setups, typical input sequence length is in the order of a few thousands. This is certainly feasible with the setup proposed by Farzi’s latent parameterization (but would struggle if the sequence length, e.g., exceed hundreds of thousands). Further, performing the Farzi optimization for larger models is also possible out-of-the-box with hardware/compute requirement no-more-than training the LLM itself. The only reason we weren’t able to perform data distillation for LLMs is because we don’t have the compute to train LLMs by itself.
>
> **Q: What's the overall cost saving versus just training a model on the original dataset for more time?**
>
> A: Great question! Since this question was also raised by other reviewers, we have put the results in a separate comment up top for the sake of not repeating, and are happy to present our impressive results!
>
> **Q: Have the authors thought about cases where there is significant noise in the training corpus?**
>
> A: Wonderful question, and highly relevant in, e.g., pre-training scenarios as you mentioned. Even though we don’t have direct evidence in this paper, there is indeed very strong hope for automatic denoising while performing data distillation. Looking at Figure 4 in [1] which also performs data distillation, training on data summarized by data distillation is (1) definitely much better than training on random samples of noisy data; and (2) better than training on the full dataset when there is high-noise in the datasets. The intuition behind this is that data distillation implicitly promotes de-noising, as the overall optimization objective is to maximize model performance within a small data budget, implicitly minimizing the noise in such low-data scenarios.
>
> We would like to end by requesting you to kindly reconsider your final rating as it strictly decides the outcome of this paper, which in your own self-opinion is well-timed, and a fresh approach to data optimization for autoregressive tasks like NLP.
>
> [1] Sachdeva, Noveen, et al. "Infinite recommendation networks: A data-centric approach." Advances in Neural Information Processing Systems 35 (2022): 31292-31305.

---

### Official Review · Reviewer_FjiL · 2023-11-05

**Soundness:** 2 fair
**Presentation:** 4 excellent
**Contribution:** 2 fair
**Rating:** 5
**Confidence:** 4

**Summary:**

The paper introduces FARZI, a data distillation framework for machine learning tasks. The goal is to condense the original large dataset into a much smaller number of synthetic sequences, so that downstream performance on the synthetic data matches (or even improves) performance on the full real dataset. The authors cast the problem using a bi-level optimization formulation, similar to meta-model matching based dataset distillation. The naive formulation is infeasible due to the very large token vocabulary and the maximum sequence length. To address this, the authors propose to factorize the synthetic dataset into a latent data summary and a token-decoder matrix. This renders the optimization continuous (as opposed to discrete), while it provides flexibility to sample synthetic sentences from a distribution (as opposed to having a fixed small set of synthetic sentences). Furthermore, the authors suggest to replace SGD in the inner loop by the Adam optimizer. To mitigate the large memory footprint, they derive an efficient approximation for reverse-model differentiation of the Adam optimization. The authors assess FARZI on sequential recommendation and language modeling tasks, where they manage to match or even exceed the downstream full-data performance using as little as 0.1% of the original dataset. The authors conduct several experiments and ablation studies to shed light on various aspects of their framework.

**Strengths:**

The paper makes several interesting contributions. The meta-model matching based dataset distillation was originally proposed for continuous data (e.g., image data), as opposed to language data that use discrete tokens. The use of a latent space addresses this challenge by ensuring that the optimization can be performed in a continuous space, but by also allowing us to sample the synthetic sentences from a compact distribution. Furthermore, the observation that the Adam optimizer is a much better choice for the inner loop optimization (compared to SGD) is very interesting and dramatically improves downstream performance. To address the large memory footprint, the authors derive an efficient approximation of the reverse-mode differentiation of the Adam optimizer, which nicely complements their finding that Adam is better than SGD. Interestingly, this may be more broadly applicable in other bi-level optimization tasks (e.g., in a meta-learning context).

The paper is well written and the related work is covered quite extensively. The authors describe in detail the various insights of their framework. When it comes to the experimental evaluation, they provide a lot of information on the metrics, datasets, hyperparameters, objectives, and even architectures.

The experimental evaluation is quite convincing and supports the claims made by the authors. It is very interesting that FARZI can even outperform downstream performance on the full original dataset, which could indicate the improved robustness with dataset distillation. I liked the fact that the authors investigated various aspects of FARZI, such as the versatility of the synthetic data, the cross-architecture generalization, the performance of different meta-objectives, the cold start problem, and the impact of pre-trained trajectories.

**Weaknesses:**

1. Even though this paper makes interesting contributions to the DD literature for autoregressive tasks, it is not so obvious that it would be
very helpful for much larger text corpora and large language models with millions or billions of parameters. The memory footprint might end up being very large, rendering the whole framework infeasible. Furthermore, a compression rate of 0.1% may not be extremely helpful for very large datasets consisting of billions of sentences. This may limit the applicability of FARZI to settings consisting of "reasonably large but not very large" language corpora.

2. It was not clear to me how time-consuming the FARZI dataset generation process is. For example, how long did it take to generate the synthetic datasets for the tasks considered in this work? In particular, did FARZI improve the total runtime? For instance, if generating the synthetic data takes very long, then there may be very little benefit (if any) from this process. Furthermore, it is not automatically obvious that a smaller dataset can be trained faster than a larger one. There is the added question of the number of epochs required to reach convergence. The synthetic dataset may require more rounds. This was not obvious in the experimental evaluation. If I am not mistaken, I feel that the subject of runtime was only superficially touched in this work, and a more thorough discussion (with detailed pros and cons) would be needed.
(Theoretically, this may not be a big issue if the same synthetic dataset could be successful used on several downstream tasks, but this is not immediately true. If we need dataset distillation for each separate task, then we may end up performing FARZI several times.)

**Questions:**

1. Could the authors elaborate more on the total runtime (total time for synthetic dataset generation + total time for downstream training with synthetic vs. full data)? It would be helpful if the authors could shed light on the various questions/comments raised in Weakness (2) above.

2. In Equation (2), \Omega is a set containing initializations for the inner loop, if I understand correctly. But instead of picking the initialization randomly, these come from a small number of training trajectories on the full dataset. If that is true, then the \theta_i in the definition of \Omega has nothing to do with the update rule for \theta_t in Equation (2). This may still be confusing to some readers though because the same symbols are used (theta with a subscript, so the authors may want to clarify this point (i.e., what exactly is in \Omega).

3. I was not clear how exactly the authors chose the final hyperparameters for each setting. Did they exhaustively try all corresponding combinations in the hyperparameter table and picked the best one?

4. Is a new synthetic batch created at the beginning of each outer-loop step based on the latent factorization?

---

> ### Author Response · Authors · 2023-11-23
> **Response to reviewer FjiL**
>
> We really thank you for your time and effort in reviewing the paper and providing us with your valuable feedback! Below are the responses to your well-motivated questions, which we hope will shed more clarity on Farzi’s design. Please let us know if we can add / expand on something:
>
> **Q: It is not so obvious that Farzi would be very helpful for much larger text corpora and large language models**
>
> A: While we absolutely agree that large language models would be an exciting application for Farzi, we would like to mention that most research labs do not have the compute necessary for such line of work. Hence, we believe penalizing for such reasons (that is only if considered while rating this paper) might be a very strict requirement that implicitly undermines exploratory or academic research.
>
> **Q: Could the authors elaborate more on the total runtime.**
>
> A: Great question! Since this question was also raised by other reviewers, we have put the results in a separate comment up top for the sake of not repeating, and are happy to present our impressive results!
>
> **Q: $\theta_i$ in the definition of $\Omega$ has nothing to do with the update rule for $\theta_t$ in Equation (2)**
>
> A: Thanks for pointing this out! There is definitely a clarification to be made here: $\Omega$ contains the *flat* set of all episodic checkpoints for the training trajectories and when we sample $\theta_0$ (0 here represents the inner loop time-index), it can come from the union of all the intermediate checkpoints available to us ($\Omega$). We’ll make sure to add the clarification in the pdf as well.
>
> **Q: It was not clear how exactly the authors chose the final hyperparameters for each setting**
>
> A: Just like any deep learning setup, we test a random subset (as per a reasonable grid search budget) of the grid of combinations listed in Tables 4 & 5 and report the best result after selecting the best hyper-parameters using a validation set.
>
> **Q: Is a new synthetic batch created at the beginning of each outer-loop step based on the latent factorization?**
>
> A: Thanks for the question! We list the sequence of steps in Farzi for some added clarity and will add this as an algorithm block in the appendix:
> 1. Before any optimization we randomly initialize $\tilde{\mathcal{D}}_{syn}$ and $\mathbf{M}$
> 2. Sample a batch of fake sequences from $\tilde{\mathcal{D}}_{syn}$
> 3. Perform one outer loop step as demonstrated in figure 2
> 4. Repeat from step 2, till convergence
>
> We would like to end by requesting you to kindly reconsider your final rating as it strictly decides the outcome of this paper, which in your own self-opinion contains important technical contributions, and contains strong empirical evidence.

---

### Official Review · Reviewer_At7H · 2023-11-10

**Soundness:** 3 good
**Presentation:** 3 good
**Contribution:** 3 good
**Rating:** 6
**Confidence:** 3

**Summary:**

This paper proposes FARZI, a data distillation method for auto-regressive ML tasks/event-sequence datasets. The method summarizes a large dataset into a set of synthetic sequences in latent space which can be decoded later. They show that model performance is upheld/enhanced when compared to training on the complete dataset on the downstream tasks of sequential recommendation and language modeling. For data distillation, the paper shows Adam to be better than SGD as inner loop optimizer, and derives an efficient reverse mode differentiation of Adam such that its memory complexity is independent of the number of inner loop steps.

**Strengths:**

- Originality and Significance: The latent parametrization that makes FARZI optimization friendly, and the proposed trick that enables reverse mode differentiation of Adam such that its memory complexity is independent of the number of inner loop steps are great contributions and of practical value.
- Quality and Clarity: The paper is well written with extensive experiments whose details and evaluations are that are clearly described. The results are impressive. The method is able to achieve better performance on downstream tasks compared with using the full dataset.

**Weaknesses:**

- It is not clear whether this method will be practical and scale for larger language models and larger datasets. It would be great if the authors can elaborate on this.
- There is not a clear analysis of the total time gains of this method in comparison with training from scratch. Providing some values would make the case for this method more compelling.

**Questions:**

Listed in weakness section.

---

> ### Author Response · Authors · 2023-11-23
> **Response to reviewer At7H**
>
> We really thank you for your time and effort in reviewing the paper and providing us with your valuable feedback!
>
> Below are the responses to your important and well-needed questions, and please let us know if we can add / expand on something:
>
> **Q: It is not clear whether this method will be practical and scale for larger language models and larger datasets.**
>
> A: We split the response into two parts, one for larger datasets, and one for larger models:
> - The model is out-of-the-box scalable to as large datasets as we want, simply because the model only requires *batches of the dataset we want to summarize*. For more context, the largest dataset used in the paper (Netflix) has almost half a million data-points, and we only use a batch size of $512$ in the outer loop. Even with such an extremely small batch size, only $2,000$ sequences synthesized by Farzi can train models equally well as training on all half a million sequences in the original dataset.
> - Being restricted to an academic research environment, we don’t have the compute necessary to experiment with large language models (LLMs). Even though it would be a well-timed application to LLMs, we believe Farzi paves a clear way for future research in data distillation for LLMs. Nonetheless, we would like to point out that Farzi’s reverse-mode Adam derivation makes it possible to perform data distillation for LLMs using no more than the same hardware that is used to train such LLMs. In other words, the memory complexity for Farzi is the same as training the base model.
>
> **Q: There is not a clear analysis of the total time gains of this method.**
>
> A: Great question! Since this question was also raised by other reviewers, we have put the results in a separate comment up top for the sake of not repeating, and are happy to present our impressive results!
>
> We would like to end by requesting you to kindly reconsider your final rating as it strictly decides the outcome of this paper, which in your own self-opinion is of good practical value, and contains strong empirical evidence.

---

### Author Response · Authors · 2023-11-23
**Time benefits of using Farzi data**

We note that multiple reviewers asked for time advantages of using the Farzi optimization. We believe this is a great point that is worth talking about, and almost the entirety of the data distillation literature doesn’t focus on the wall-clock time improvements. However, since we absolutely agree with this line of thought, we present the wall-clock times of some of the major operations in Farzi (and will include it in the main-text) while summarizing the largest dataset used in our experiments (Netflix dataset, 476k sequences), all using the same codebase and hardware environments:

| Data size | Operation | Time |
|----------|:-------------:|------:|
$[2000 \times 200]$ | Farzi optimization (fake data optimization) | 7500s (15s per 500 outer loop steps) |
$[2000 \times 200]$ | Model training on fake data | 172s total, optimal at 100 epochs ($\equiv 0.2$M sequences) |
476k sequences | Model training on entire dataset | 3700s total, optimal at 30 epochs ($\equiv 14$M sequences) |

Please note that, as other reviewers have pointed out, the cost of performing the fake data optimization can easily be amortized in very practical scenarios where we need to train multiple models (or even perform, e.g., hyper-parameter tuning) on the same dataset, which is almost always the case. Such amortization strictly depends on the ratio of the aforementioned quantities in the table, and in our case, the cost of data optimization is only $2\times$ the cost of training a model on the entire dataset. This is a very reasonable amortization trade-off for more than an order of magnitude future training speed-up (172s vs. 3700s).

---

### Meta-Review · Area_Chair_hkYR · 2023-12-16

**Metareview:**

This paper proposes a new algorithm for distilling sequential datasets into smaller synthetic sequences. The goal is to improve and accelerate the training of autoregressive generative models by converting the original data into a small, structured 3d summary tensor that mixes the interactions across different sequences in a compressed form. The approach follows meta-learning with the outer loop updates the summary tensor and the inner loop optimizes downstream model performance on a specific data summary. While similar ideas have been explored for vision, the reviewers appreciated the introduction of a meta-learning approach for language modeling which presents unique challenges due to the discreteness and variations in token lengths. The experiments also showed good gains at the tested scales. The reviewers pointed out two major concerns with the current work: first, the reviewers pointed out a fatal flaw in the theory which was a key motivation for the work; second, the large wall-clock times for data distillation beg a deeper empirical study into amortizing this cost over training of multiple models with the distilled dataset. Unfortunately, while the authors acknowledged the reviewer concerns, the reviewers commented on a lack of initiative on part of authors to make the necessary amends in the paper pdf. I would highly recommend the authors to address these issues and aim for a more comprehensive and balanced overview of their approach in a revised version of the paper.

**Justification For Why Not Higher Score:**

Fatal flaw in theory. Empirical justification for amortizing high wall-clock time missing.

**Justification For Why Not Lower Score:**

N/A

---

### Decision · Program_Chairs · 2024-01-16

Reject